# Pretrained transformers applied to clinical studies improve predictions of treatment efficacy and associated biomarkers

Gustavo Arango-Argoty[1] ✉, Elly Kipkogei[1], Ross Stewart[2], Gerald J. Sun[1], Arijit Patra[3], Ioannis Kagiampakis[1] & Etai Jacob ®[1] ✉

Cancer treatment has made significant advancements in recent decades, however many patients still experience treatment failure or resistance. Attempts to identify determinants of response have been hampered by a lack of tools that simultaneously accommodate smaller datasets, sparse or missing measurements, multimodal clinicogenomic data, and that can be interpreted to extract biological or clinical insights. We introduce the Clinical Transformer, an explainable transformer-based deep-learning framework that addresses these challenges. Our framework maximizes data via self-supervised, gradual, and transfer learning, and yields survival predictions surpassing performance of state-of-the-art methods across diverse, independent datasets. The framework's generative capability enables in silico perturbation experiments to test counterfactual hypotheses. By perturbing immune-associated features in immunotherapy-naive patients, we identify a patient subset that may benefit from immunotherapy, and we validate this finding across three independent immunotherapy-treated cohorts. We anticipate our work will empower the scientific community to further harness data for the benefit of patients.

Over the last decade, immunotherapies have transformed cancer, however, it remains difficult to identify patients who will best respond to these therapies. Anti–PD-1/L1, anti–CTLA4, and other checkpoint blockade approaches do not directly target the tumor but instead recruit the patient's immune system to fight the disease. Due to this indirect mechanism of action, the drivers of response to therapy are more complicated than those of, for example, a targeted inhibitor of an oncogene. Response can be affected by the patient's physical condition and immune system fitness, as well as the tumor's underlying biology. Thus, despite advancements in precision medicine, biomarker discovery[1,2], and machine learning–based modeling[3–5], predicting patient response and understanding mechanisms of resistance to these therapies remain challenging[6,7]. At present, only three predictive biomarkers–the assessment of microsatellite instability (MSI), tumor PD-L1 expression by immunohistochemistry, and the measurement of

tumor mutation burden (TMB)[8–10]–have received approval from the U.S. Food and Drug Administration (FDA) for use as companion diagnostics for immunotherapy.

Recently, deep-learning artificial intelligence (AI) transformer models have gained widespread adoption in advanced natural language modeling applications[11], including ChatGPT[12,13], image processing[14,15] (e.g. Dino[16], ViT[17]), protein structure prediction (e.g., AlphaFold)[18,19], and de novo protein sequence generation[20,21]. A key feature of transformers is self-attention, a mechanism designed to identify dependencies among features[22]. Self-attention is crucial in language processing tasks, where the meaning of a word depends on its context within a sentence. Similarly, in precision medicine, a biomarker's significance may be limited if considered in isolation from other clinical or molecular features. Transformers can evaluate the importance of various disease biomarkers within the context of all

[1]Oncology Data Science, Oncology R&D, AstraZeneca, Waltham, MA, USA. [2]Translational Medicine, Oncology R&D, AstraZeneca, Cambridge, UK. [3]Clinical Pharmacology & Safety Sciences, BioPharmaceuticals R&D, AstraZeneca, Cambridge, UK. ✉e-mail: gustavo.arango@astrazeneca.com; etai.jacob@astrazeneca.com

available clinical and molecular data, dynamically adjusting their influence on outcomes such as patient response to treatment. Therefore, the ability of attention mechanisms to potentially capture complex relationships between patient characteristics and response, as demonstrated in applications like natural language processing, could improve outcome predictions and help identify the best treatment for a patient.

However, transformer models for clinically translatable use need to meet certain criteria that differ from those of other applications. First, models must be compatible with relatively small datasets, such as those in clinical studies (e.g., 100 or fewer to 1000 patients in phase 1–3 clinical trial cohorts), in contrast to training data for imaging and natural language applications. Second, models must be able to adeptly manage sparse features, such as functional mutation events from genomic profiling, that are typically infrequent within a patient population. Third, predictive models require incorporation of multiple types of clinical and real-world data, including but not limited to, features derived from DNA and RNA sequencing from tumor or peripheral blood. In addition, some measurements could be missing in more than one part of a patient cohort (e.g. due to unsuccessful assays or biopsies or limited measurements). In addition to these challenges, biological and clinical data are noisy, variable, and inconsistent (e.g. batch effects or data generated by different assays or laboratories). Therefore, models for clinical applications must be able to effectively integrate high-dimensional and diverse data, combine patient populations, and handle missing data. Fourth, deep-learning models are often considered "black boxes" because they provide predictions without explaining how they were derived. However, models used in a clinical domain need to be interpretable. This allows researchers to verify alignment with existing knowledge, relate clinical and molecular features to survival predictions, increase confidence in these predictions, and connect results to clinical contexts, such as associating survival predictions with features that support certain treatments. Finally, extracting biological insights and understanding disease biology from prediction models in the relevant clinical context is vital for effectively informing further research. This involves providing a mechanistic understanding of drug effects, identifying potential molecular targets, and elucidating mechanisms of resistance to treatment.

To address these needs, we created the Clinical Transformer, a deep neural-network survival prediction framework based on transformers. This framework effectively handles relatively small datasets through a transfer learning mechanism. It can be used to build foundation models by leveraging large datasets (e.g., The Cancer Genome Atlas [TCGA] and the Genomics Evidence Neoplasia Information Exchange [GENIE]), followed by task-specific fine-tuning with smaller datasets (e.g., a patient cohort from a clinical study). We show that the Clinical Transformer exhibits flexibility in handling diverse datasets with various feature types and levels of sparsity, as well as the ability to combine patient populations. Owing to its attention mechanism, the Clinical Transformer captures complex, nonlinear relationships among multiple molecular, clinical, and demographic patient features. Across 12 cancer datasets comprising more than 140,000 patients, and including individuals treated with immuno-oncologic (IO), targeted, and chemotherapy treatments, the Clinical Transformer consistently outperforms state-of-the-art methods used for survival prediction, including Cox proportional hazards (CoxPH) and random survival forest. We describe how the internal representations learned by the fully trained model could be used for specialized tasks, such as predicting immunotherapy response in small real-world datasets or early-stage clinical trials. Finally, we show how the Clinical Transformer's explainability and perturbation modules improve clinical interpretations of outcome and response predictions by identifying molecular and clinical features that lead to specific outcomes in individual patients.

## Results

### The Clinical Transformer framework

The Clinical Transformer framework (Fig. 1a) consists of: input-agnostic modeling for integrating multiple molecular (e.g., omics), clinical, and demographic modalities into a single system, handling sparse features, missing data, and different annotation hierarchies (e.g., gene vs. protein name) (Fig. 1b); self-supervision and transfer learning for analysis of small datasets typical in clinical studies (Fig. 1c); an interpretability module that can suggest biological insights for clinical applications that require high confidence (Fig. 1a); and a generative model for creating synthetic populations of patients by modifying real patient features and predicting virtual responses. This generative AI capability enables the exploration of scenarios that affect resistance or enhance response across a diverse array of patient populations (Fig. 1a).

The Clinical Transformer employs three learning strategies: (1) direct learning, in which the model is trained from scratch for a task like survival or response prediction; (2) gradual learning, in which, using the same input data for survival prediction, the model is first trained with self-supervised learning for masked feature prediction, similar to the strategy used to train large language models such as BERT[23] (Fig. 1c), and is then fine-tuned for specific tasks (e.g., patient response or survival prediction); and (3) transfer learning, in which a model is pretrained on large amounts of data, using self-supervision for masked feature prediction, and the weights are then used to initialize other models that are fine-tuned to predict survival or patient response. The pretraining step (i.e., gradual learning and/or transfer learning) enables the model to identify general biological and clinical patterns in the data, offering a more informative starting point than random initialization, and allows further refinement of a new model that focuses on patient outcome as a target function (Fig. 1c).

### Performance compared with state-of-the-art survival prediction models

In total, 12 datasets from clinical trials and real-world data (pan- and cancer-specific), totaling 156,192 patients, were used for multiple tasks, including direct, gradual, and transfer learning (Table 1, Supplementary fig. 1, Methods). We evaluated the Clinical Transformer against common methods like other neural network-based methods[24–28], CoxPH, TMB as risk score, and random survival forest, using five independent datasets (Table 2).

In a pan-cancer setting evaluated on a study by Chowell et al.[3], which integrated multiple variables into a machine learning model to predict patient response to immunotherapy[3], the Clinical Transformer achieved a concordance index (C-index)[29] of 0.73, outperforming the Chowell et al.[3] random forest model (0.68) and TMB (0.55; recently FDA-approved for this purpose[30]; Fig. 2a–c). Models were trained using the same training and testing data splits as in the Chowell study, along with the reported prediction scores from the trained random forest model and TMB (Supplementary Note 1).

To assess the Clinical Transformer's ability to stratify patients, we classified patients into high- or low-risk populations using specific cutoffs: 0.239 for the Chowell et al.[3] classifier (the optimal cutoff that study reported)[3]; 10 mutations per megabase (mut/mb) for TMB (the FDA-approved cutoff)[30]; and the median score derived from the training set for our model. The Clinical Transformer best stratified patients (Fig. 2a), with a hazard ratio (HR) of 0.29 (95% confidence interval [CI]: 0.21–0.40; $P$ = 3e-14), outperforming the Chowell et al.[3] random forest model (HR: 0.34, 95% CI: 0.25–0.47; $P$ = 2e-11), and TMB (HR: 0.69, 95% CI: 0.50–0.97; $P$ = 3e-2) (Fig. 2b, c).

To evaluate the gradual learning strategy, we pretrained the Clinical Transformer in a self-supervision mode for 30,000 iterations on the complete Chowell et al.[3] dataset and then fine-tuned it on the dataset's survival endpoints. We then evaluated its performance on an independent dataset of 150 patients with non–small-cell lung cancer

**Fig. 1 | Overview of the Clinical Transformer framework. a** Framework capabilities and analysis overview of the Clinical Transformer to process data to insights. Model interpretability is extracted by using the output embeddings to generate functional modules (group or input features) that are associated with the outcome. The Clinical Transformer is shown as a generative model to recreate a patient's trajectory of response, using patient embeddings with a perturbation-based approach. Embeddings can also generate synthetic data restricted by certain conditions. **b** Input data are represented by [Key, Value] pairs, where the key is the feature name and the value corresponds to the numerical score of the feature (e.g., [Age, 20] represents a patient with an age of 20 years). Feature names and values are embedded and fed to a transformer encoder architecture without positional encoding. The special input token [TASK, 1] is added in front of every input sample, and the output of this token is used to predict patient survival or classification

outcomes. The special token [MASK] is used for performing the pretraining stage, in which the model is asked to predict the masked token names (corresponding to the masked features). **c** The Clinical Transformer trained in self-supervised mode uses dataset A to train over a masked-prediction task in which input features are randomly ignored and used as labels. Thus, the objective of the model is to predict the feature name of the ignored input features. After pretraining, the weights of the model can be used to fine-tune into a specialized task such as responder prediction or survival analysis. When input data A are different from input data B, it refers to transfer learning, whereas if the same data are used for both tasks (data A = data B), it is called gradual learning, as the model first learns about the data in an unsupervised way and then specializes on a specific task over the same dataset (e.g., survival prediction).

(74 patients treated with anti-PD-L1 and 76 treated with anti-PD-L1 + anti-CTLA-4 from the MYSTIC trial [NCT02453282]; Supplementary Note 2). The Clinical Transformer showed superior performance with gradual learning (C-index: 0.643; HR: 0.50, 95% CI, 0.34−0.74; $P = 2.64e\text{-}3$) compared to direct learning (C-index: 0.616, HR: 0.56, 95% CI, 0.38−0.81; $P = 3.79e\text{-}3$) and TMB (C-index: 0.608, HR: 0.68, 95% CI, 0.47−0.99; $P = 4.4e\text{-}2$) when trained on the Chowell et al. data and evaluated on MYSTIC data (Supplementary fig. 2, Supplementary Table 1). These results indicate the benefit of gradual learning for predicting survival outcomes with limited datasets.

The Clinical Transformer outperformed all other approaches in independent evaluations of IO treatment arms in the MYSTIC[31] study (anti-PD-L1, anti−PD-L1 + anti−CTLA-4; NCT02453282) and OAK[32] trials (anti−PD-1/PD-L1; NCT02008227). It achieved survival predictions with C-indexes of 0.67 and 0.669 for MYSTIC and OAK, respectively, compared to the random survival forest model (C-indexes: 0.606 and 0.664), CoxPH (C-indexes: 0.599 and 0.620), and TMB (C-index: 0.589 for MYSTIC) (Table 2).

We next evaluated the Clinical Transformer's performance using nonclinical features. Using data from Samstein et al.[33], a molecularly-derived study which profiled 1662 tumors from 32 different cancer types using the Memorial Sloan Kettering (MSK) Integrated Mutation Profiling of Actionable Cancer Targets (IMPACT) panel

(Supplementary Note 3), the Clinical Transformer achieved a C-index of 0.649, surpassing the random survival forest (0.638), CoxPH (0.594), and TMB (0.543). Using immunogenomic features of the tumor microenvironment (TME), reported by Thorsson et al.[34] (Supplementary Note 4), the Clinical Transformer showed improved prediction of overall survival (OS) and similarly outperformed the random survival forest, CoxPH, and TMB models (Table 2).

Overall, the Clinical Transformer consistently outperformed state-of-the-art methods, including other neural network and transformer-based models, in predicting patient survival (Table 2).

**Pretraining with large, unlabeled real-world data to predict patient response in small immunotherapy cohorts**

"Real-world data" broadly refers to data generated during routine clinical practice and includes datasets such as UK Biobank[35], Flatiron[36], TEMPUS[37], and GENIE[38]. GENIE, a cancer registry from 19 leading international cancer centers, links clinical and genomic data to support the identification of biomarkers associated with treatment response. We incorporated the use of transfer learning to take advantage of GENIE data. Similar to large language models (e.g., ChatGPT), which are pretrained on large amounts of data for specific language-related tasks, the Clinical Transformer uses unlabeled real-world data to learn general patterns, which are then fine-tuned (or specialized) for tasks

**Table 1 | Datasets used in this study**

| Dataset (reference) | Cancer type | Description | Data usage | No. of patients | Treatment | No. of features | Features |
|---|---|---|---|---|---|---|---|
| Chowell et al.[3] | Pan-cancer | Patients treated with anti-PD-1/PD-L1 and combo in multiple cancer types | Survival modeling (training and testing), transfer learning | 1479 | IO | 17 | Dataset consisting of 17 features (e.g., TMB, HLA-LOH, MSI, albumin, NLR) |
| MYSTIC[31] trial (NCT02453282) | NSCLC | Stage IV NSCLC treatment naive anti-PD-L1 and anti-PD-L1 + anti-CTLA-4 | Survival modeling (training and testing) | 325 | IO | 15 | Dataset consisting of 15 features (e.g., TMB, HLA-LOH, MSI, albumin, NLR) in addition to FMI mutation calls, in addition to a subset of 150 with HLA germline typing |
| OAK[32] trial (NCT02008227) | NSCLC | Stage IV NSCLC patients treated with anti-PD-1/PD-L1 after failure of chemotherapy | Survival modeling (training and testing) | 396 | IO | 418 | Including patients with available ctDNA-derived mutation calls |
| Samstein et al.[33] | Pan-cancer | Response to IO treatment (anti-PD-1/PD-L1 and combo) in pan-cancer setting | Survival modeling (training and testing) | 1610 | IO | 474 | Tissue MSK-IMPACT calls from 474 genes in addition to demographic features |
| Thorsson et al.[34] | Pan-cancer | Study on TCGA data that derives features associated with the tumor micro environment in a pan-cancer setting | Survival modeling (training and testing) | 6012 | Untreated | 49 | A total of 49 derived features from RNA sequencing data that profile the TME |
| Bagaev et al.[80] | Pan-cancer | TME subtypes derived from TCGA data | Pretraining and survival modeling (training and testing) for SKCM | 11070 | Untreated | 29 | A total of 29 RNA signatures obtained from this study |
| Van Allen et al.[74] – DFCI metastatic melanoma | Melanoma | Metastatic melanoma dataset from DFCI; anti-CTLA-4 monotherapy | Validation dataset | 110 | IO | 36,859 | Whole exomes from pretreatment melanoma tumor biopsies and matching germline tissue samples from 110 patients. 40 patients with RNA expression |
| AACR project GENIE (Pugh et al.[38]) | Pan-cancer | Publicly accessible international cancer registry of real-world data assembled through data sharing among 19 of the leading cancer centers in the world | Transfer learning | 134,626 | Not available | 2290 | Mutation and copy number calls from multiple panels and centers spanning 2290 genes in addition to demographic features |
| MSK MIND (Vanguri et al.[4]) | Lung | Initiative to accelerate research and discovery through advanced analytics | Validation dataset | 247 | IO | 1043 | Cohort of 247 patients with advanced NSCLC with multimodal baseline data; only molecular data used |
| Miao et al.[73] | Pan-cancer | WES of 249 tumors from patients with clinically annotated outcomes to immune checkpoint therapy | Validation dataset | 249 | IO | >10,000 | WES and demographic features |
| Riaz et al.[82] | Melanoma | Whole transcriptome from patients who progressed on ipilimumab or were ipilimumab naive, before and after nivolumab | Validation dataset | 26 | IO | 29 | Extracted 29 signatures defined by the Bagaev et al.[80] signatures using whole transcriptome data. Included baseline samples of patients that progressed on ipilimumab |
| Liu et al.[81] | Melanoma | Whole transcriptome for patients treated with anti-PD-1 therapy | Validation dataset | 42 | IO | 29 | Extracted 29 signatures defined by the Bagaev et al.[80] signatures using whole transcriptome data. Including samples acquired before anti-PD-1 treatment from patients receiving prior treatments to anti-PD-1 |

Boldface indicates column header.
ctDNA circulating tumor DNA, DFCI Dana Farber Cancer Institute, FMI Foundation Medicine, WES whole-exome sequencing.

**Table 2 | Overall performance of the Clinical Transformer compared with other modeling approaches. Results are reported as mean c-index ± standard deviation across the 10 train/test splits**

| Modeling framework | Learning strategy | MYSTIC trial (NSCLC) | OAK trial (NSCLC) | Samstein et al.[33] pan-cancer | Thorsson et al.[34] pan-cancer (TCGA) | Chowell et al.[3] pan-cancer | Chowell et al.[3] evaluated on MYSTIC |
|---|---|---|---|---|---|---|---|
| Clinical Transformer | Neural network | **0.670 ± 0.07**[a] | **0.669 ± 0.04** | **0.649 ± 0.02**[b] | **0.734 ± 0.01** | **0.720 ± 0.01** | **0.643 ± 0.004**[a] |
| Cox-nnet | Neural network | 0.609 ± 0.04 | 0.626 ± 0.04 | 0.622 ± 0.02 | 0.676 ± 0.01 | 0.707 ± 0.01 | c |
| DeepSurv | Neural network | 0.602 ± 0.05 | 0.620 ± 0.04 | 0.595 ± 0.02 | 0.707 ± 0.01 | 0.691 ± 0.01 | c |
| Neural MTLR | Neural network | 0.626 ± 0.05 | 0.577 ± 0.03 | 0.538 ± 0.02 | 0.658 ± 0.05 | 0.680 ± 0.01 | c |
| Nnet-survival | Neural network | 0.603 ± 0.04 | 0.563 ± 0.03 | 0.527 ± 0.02 | 0.691 ± 0.01 | 0.695 ± 0.02 | c |
| Transformer Survival | Neural network | 0.584 ± 0.03 | 0.583 ± 0.03 | 0.608 ± 0.02 | 0.706 ± 0.006 | 0.709 ± 0.01 | c |
| Linear modeling | CoxPH regression | 0.599 ± 0.03 | 0.620 ± 0.03 | 0.594 ± 0.03 | 0.690 ± 0.01 | 0.709 ± 0.01 | c |
| Nonlinear modeling | Random survival forest | 0.606 ± 0.04 | 0.664 ± 0.05 | 0.638 ± 0.01 | 0.722 ± 0.01 | 0.714 ± 0.01 | c |
| Biomarkers | TMB | 0.589 ± 0.05 | 0.543 ± 0.05 | 0.538 ± 0.02 | 0.624 ± 0.01 | 0.550 ± 0.02 | c |

Boldface indicates column header or modeling framework with best performance.
[a]Chowell et al. pretraining.
[b]GENIE pretraining.
[c]Not evaluated.

like predicting treatment response or survival in smaller clinical trial datasets.

We implemented pretraining and transfer learning using GENIE data and fine-tuned on four, smaller independent datasets (Samstein et al., MSK Multi-modal Integration of Data (MIND), MYSTIC, and Dana Farber Cancer Institute [DFCI] datasets; Table 1) to predict patient survival (Supplementary Note 5). Transfer learning based on GENIE improved survival predictions across all datasets (average C-index, 0.617) compared to direct learning (average C-index, 0.583) (Supplementary Table 2), and, on average, reduced training time by 40% (Fig. 2d, e; Supplementary fig. 3). We further evaluated the advantage of transfer learning by using a different independent dataset, from Chowell et al.[3], for pretraining, followed by fine-tuning on MYSTIC trial data (Fig. 2f; Supplementary Note 6). This achieved a C-index of 0.670 versus 0.628 for direct learning (Mann-Whitney-Wilcoxon test, two-sided; $P = 0.045$). For patient stratification, transfer learning achieved an HR of 0.49 (95% CI: 0.21–1.15; cutoff: average median risk) across 10 testing splits, compared to an HR of 0.57 (95% CI, 0.24–1.33) without transfer learning. Superior performance via pretraining and transfer learning was also achieved on other tasks than survival such as predicting primary versus metastatic sample origin (Supplementary Note 7; Supplementary Table 3). Together, these analyses underscore the versatility of the Clinical Transformer and its potential to build foundation models adaptable to various downstream clinical tasks.

## Using the Clinical Transformer's explainability module to identify features associated with survival outcomes

Clinical applications require not only prediction results, such as risk or response prediction, but also insight into the features driving these results and their underlying assumptions[39]. Explainability is important for verifying predictions, increasing human confidence in results, and uncovering the mechanisms behind both successful and erroneous predictions, and has been an important area of research in recent years[40–42]. The Clinical Transformer's explainability module enables a comprehensive understanding of the factors influencing disease biology and their relationships with patient outcomes. To estimate feature importance in the model, we used a feature permutation importance algorithm (Methods). In Chowell et al.[3] data, albumin, neutrophil-to-lymphocytes ratio (NLR), prior chemotherapy, TMB, fraction of copy number alterations (FCNA), and hemoglobin (HGB) were the most informative features. In contrast, human leukocyte antigen (HLA) evolutionary divergence (HED), age, sex, and cancer type did not have a strong effect on the model's outcome (Fig. 3a). The Clinical Transformer's rankings of feature importance strongly aligned with those reported in the original study[3].

To understand how input features relate to immunotherapy response, we stratified patients into four categories based on quartile cutoffs of the model's predicted survival scores (Fig. 3b): (1) short-term survivors (< 25th percentile, median OS: 4.4 months); (2) mid-low survivors (25th–50th percentile, median OS: 10.3 months); (3) mid-high survivors (50th–75th percentiles, median OS: 19 months); and (4) long-term survivors (>75th percentile, median OS: 37 months). Significant differences in albumin and NLR scores were observed between short- and long-term survivors ($P < $ 1e-16). Short-term survivors displayed lower albumin scores ($\mu = 3.2$ g/L, $\delta = 0.4$ g/L) than long-term survivors ($\mu = 4.1$ g/L, $\delta = 0.27$ g/L). Short-term survivors also exhibited higher average NLR levels ($\mu = 12$, $\delta = 11.8$) than long-term survivors ($\mu = 3.15$, $\delta = 1.79$) (Fig. 3c). The high NLR variance in the short-term survivors suggests that this group may be less stable than the long-term survivor population. Interestingly, the TMB score exhibited statistical significance only when the long-term survivors were compared with all other populations ($P < $ 1e-4). However, no significant differences were observed between short-term survivors and intermediate-term survivors ($P = 0.2$) or long-term survivors ($P = 1.0$). Therefore, in accordance with the literature[33,43], high TMB scores (e.g. ≥10 mut/mb)

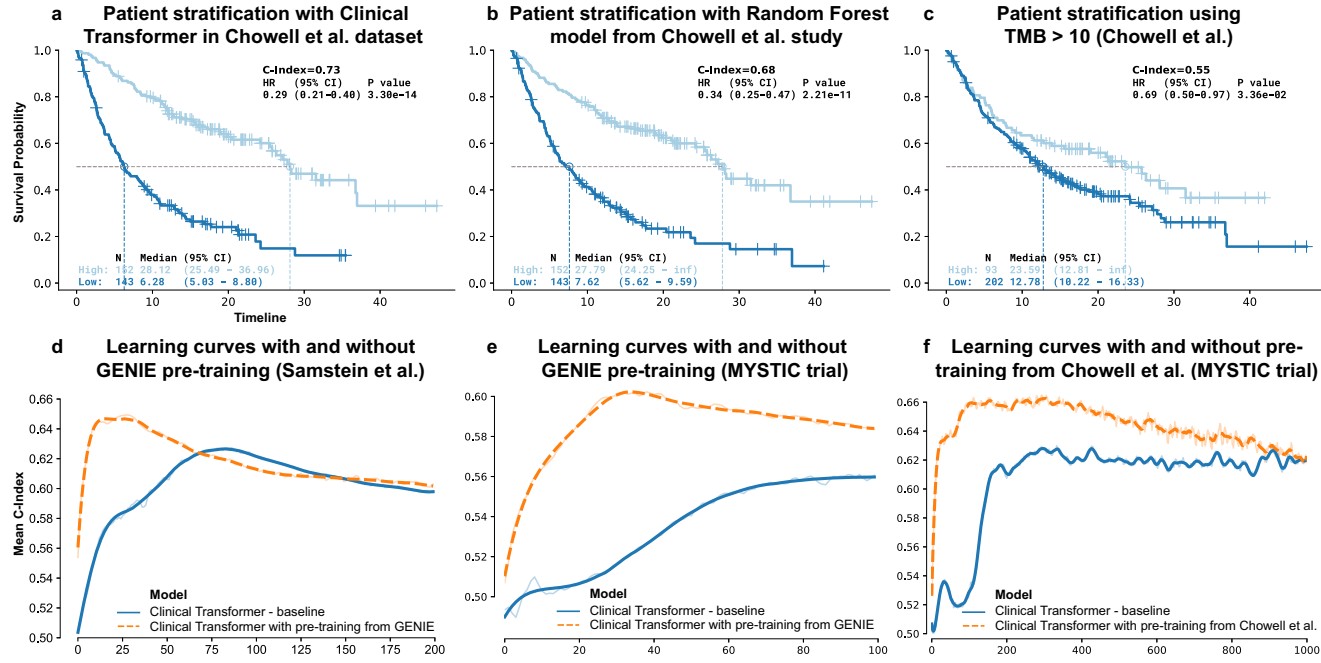

**Fig. 2 | Clinical Transformer performance and impact of pretraining. a** Kaplan-Meier (KM) curves of the Clinical Transformer applied to the Chowell et al.[3] dataset, using the median survival score to stratify patients into high and low populations. **b** KM curves of Chowell et al.[3] random forest model used in the testing dataset (using Chowell et al. optimal pan-cancer cutoff = 0.238). **c** KM curves for evaluating TMB score in the Chowell et al.[3] dataset (cutoff = 10 mutations per megabase [mut/mb]). *P*-values for HR in (**a**–**c**) reported from a Wald statistical test. **d** Learning curves of the Clinical Transformer (C-index vs. training epochs), evaluated on the 10 testing splits from the Samstein et al.[33] pan-cancer dataset, with and without using GENIE data for model pretraining. **e** Learning curves of the Clinical Transformer, evaluated on the 10 testing splits from the MYSTIC dataset, with and without using GENIE data for model pretraining. **f** Learning curves of the Clinical Transformer, evaluated on the 10 testing splits from the MYSTIC dataset, with and without using the Chowell et al.[3] dataset for model pretraining. Source data are provided in the SourceData file.

could be indicative of a positive IO treatment response, while lower TMB scores may be less informative for predicting IO responses. Moreover, in this dataset, the Clinical Transformer identified that approximately 60% of long-term survivors did not receive chemotherapy prior to IO treatment (*P* < 1e-4; Fig. 3c). Nevertheless, the generally better outcomes of patients treated with first-line therapies make it unclear if this association is specific to IO response.

## Using the Clinical Transformer's explainability module to identify key functional groups associated with survival outcomes

The attention mechanism is a key feature of the Clinical Transformer that distinguishes it from other deep neural-network architectures (e.g., fully connected and convolutional neural networks) and enables the model to identify higher-order patterns in the data, such as interactions between two features that are associated with patient survival and response. While the attention mechanism has been previously employed to interpret model predictions[44–46], previous studies considered only the attention weights, which can be influenced by nonlinear relationships within other transformer components (e.g., feed-forward network following multi-head attention, layer normalization) and do not fully capture the underlying significance of the features[47–49]. We implemented a more direct approach for assessing interpretability by calculating the cosine similarity between the model's feature and outcome embeddings, which are downstream of the transformer's neural-network components and encode both linear and nonlinear relationships between input features and clinical outcomes (Fig. 1b). Cosine similarity quantifies the relationship type and strength between two features: high positive value suggests redundancy, near-zero value suggests independence and/or orthogonality, and negative value suggests inverse relationships. This approach leverages information flow

through the entire transformer network instead of relying on scores derived from individual attention heads in the model.

In the Chowell et al[3]. dataset, albumin and TMB were the most orthogonal features (cosine similarity of 0.09; Supplementary Table 4, Supplementary fig. 4), suggesting they contribute complementary information to survival prediction. This aligns with previous studies showing that albumin enhances immunotherapy response predictions when combined with TMB[50]. We then grouped the Chowell input features into four "functional groups" based on their cosine similarity (Methods), where a group contains interacting features that share similar information related to survival outcomes (Fig. 3d):

Cluster 1 included a grouping of TMB, MSI score, and HLA−loss of heterozygosity (LOH), with a mean cosine similarity of 0.65 (0.59, 0.68, and 0.70 cosine similarities for MSI-TMB, HLA-LOH−MSI, and HLA-LOH−TMB interactions, respectively; Supplementary Table 4), implying that these three features may share a common underlying mechanism. Both MSI and TMB have been linked to improved outcomes with immunotherapy and anti−PD-1 therapy has been tumor-agnostically approved in patients with either of these markers[9,51]. Conversely, HLA-LOH has been previously associated with resistance to immunotherapy[52]. The well-documented association between high MSI and high TMB[53–56] supports the grouping of these features. A potential connection between these two features and HLA-LOH−previously shown to have a non-linear relationship with TMB[52]−is the process of antigen availability and presentation. TMB and MSI are both surrogates for neoantigen availability, and tumors with higher neoantigen loads face greater pressure to evade immune recognition through the loss of HLA.

Cluster 2 included patient-related variables, such as albumin, body mass index, HED, HGB, NLR, platelets, and age. These markers primarily reflect overall health and inflammatory status and have

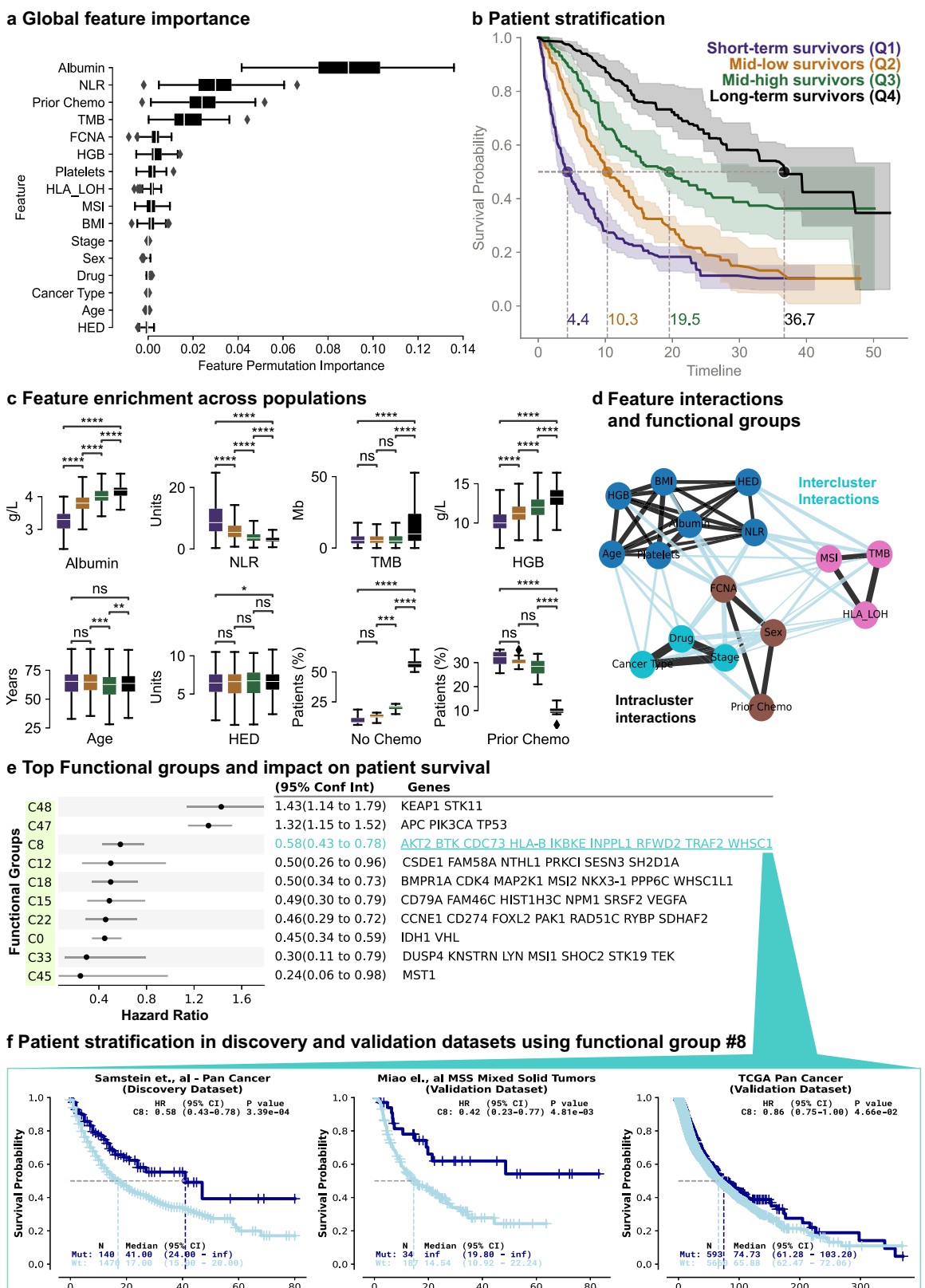

**a** Global feature importance

**b** Patient stratification

**c** Feature enrichment across populations

**d** Feature interactions and functional groups

**e** Top Functional groups and impact on patient survival

**f** Patient stratification in discovery and validation datasets using functional group #8

been associated with differential benefits from immunotherapy, though it remains unclear if they are predictive versus prognostic[57–60]. Interestingly, HED, a molecular-based biomarker derived from germline DNA, clustered with clinical laboratory markers rather than tumor-derived molecular features like TMB, MSI, or HLA-LOH. This distinction may arise because HED is derived from germline DNA and represents overall immune health and functionality, associating with overall risk of cancer and protection from infection[61,62]. Cluster 3 included FCNA, prior chemotherapy, and sex, with a mean cosine similarity of 0.60. Finally, cluster 4, which included cancer type, drug class, and tumor stage, had the highest average within-cluster cosine similarity (cosine, >0.95), perhaps

**Fig. 3 | Patient stratification and model interpretability. a** Global feature permutation importance from the Clinical Transformer applied to the Chowell et al.[3] dataset. $N = 10$ permutation test data splits. **b** KM curves (on 80% train splits) for different population groups defined by four quantile cutoffs from Clinical Transformer survival scores (Q1: $n = 241$, Q2: $n = 304$, Q3: $n = 330$, Q4: $n = 303$). Dotted lines: median survival time and probability from the 10 models on the train splits. **c** Raw feature value enrichment in the four populations used to stratify the patients (from the 10 models on 20% test splits; $n = 296$). Numerical features: y-axis in units of each variable; binary features: y-axis is proportion of patients. Bonferroni-corrected two-sided $t$ test $P$ value: ns, $5.00e\text{-}02 < P \leq 1.00e + 00$; *, $1.00e\text{-}02 < P \leq 5.00e\text{-}02$; **, $1.00e\text{-}03 < P \leq 1.00e\text{-}02$; ***, $1.00e\text{-}04 < P \leq 1.00e\text{-}03$; ****, $P \leq 1.00e\text{-}04$. **d**, Feature interaction graph derived from cosine interaction scores. Each color depicts one of the four functional groups identified when clustering the feature pairwise cosine similarities in the test sets. Black lines: intracluster interactions; light blue lines: intercluster interactions; line thickness: cosine similarity magnitude. **e** CoxPH HR (error bars: 95% CI) stratifying by each of the top 10 functional groups in the Samstein et al.[33] dataset. Functional group displayed genes capped at 10 for visualization purposes. **f** KM curves of patients possessing at least 1 mutation in group C8 (Mut) vs. no mutation (Wt), in the discovery dataset (Samstein et al.), and two independent validation datasets (Miao et al.[73] and TCGA). Boxplots in (**a**, **c**): centerline, median; box limits, quartile 1 and 3; box whiskers, 1.5× interquartile range; diamonds, outliers. Shaded region or error bars in (**b**, **e**): 95% confidence interval. HR $p$-values in (**f**) from Wald statistical test. Chemo, chemotherapy; MSS, microsatellite stable. Source data are provided in the SourceData file.

reflecting the underlying dependencies among these features in clinical practice.

We then applied this approach to the Samstein et al. dataset[33], clustering tumor tissue gene-level mutational features from the MSK IMPACT panel (469 genes; Supplementary Note 8) into 50 functional groups based on cosine similarity. The mutational features were ranked by their association with immunotherapy benefit (Fig. 3e, Supplementary fig. 5). We aggregated each functional group of multiple genes into a single binary variable: if any gene in a group was mutated, the group was assigned the value 1; if none were mutated, it was assigned 0.

Two functional groups, denoted C47 (HR = 1.32; *APC*, *PIK3CA*, and *TP53*) and C48 (HR = 1.43; *KEAP1* and *STK11*), were associated with short-term survival with immunotherapy. These findings align with prior studies showing the negative prognostic impact of *KEAP1* and *STK11* for both immunotherapy and chemotherapy[63–66]. While the impact of mutations in *APC*, *PIK3CA* and *TP53* are less consistent[67–72], these genes are key drivers of several tumors, with their impact likely varying depending on the specific setting and therapeutic regimen.

Functional group C8 contained *AKT2*, *BTK*, *CDC73*, *HLA-B*, *IKBKE*, *INPPL1*, *RFWD2*, *TRAF2*, and *WHSC1* and was associated with regulation of the immune response (GO:0002682, $P = 8.69e\text{-}4$). We used this group to stratify patients across independent IO-treated datasets and found a significant and selective benefit of IO treatment in both Samstein et al[33]. and Miao et al[73]. pan-cancer IO-treated datasets (HR = 0.58, $P = 3.3e\text{-}4$; HR = 0.42, $P = 4.8e\text{-}2$, respectively), whereas no meaningful stratification was observed for pan-cancer non–IO-treated patients in the TCGA dataset (HR = 0.86, $P = 4.6e\text{-}2$; Fig. 3f).

### Translating the Clinical Transformer to simple and interpretable linear models

We explored whether these functional groups could be leveraged to construct a simple linear model. Binary variables representing functional groups were used as inputs to a CoxPH model to predict patient OS. Multiple functional groups could be used to construct a multivariate CoxPH model.

To test this approach, we selected the top 10 molecularly derived functional groups from the Samstein et al[33]. dataset, which showed the strongest relationship with OS (Fig. 3e), as inputs to a multivariate CoxPH model to predict patient OS (Methods). The CoxPH model achieved an average C-index of 0.63 on held-out test splits from Samstein et al. data (Fig. 4a), with an HR of 0.53 on the Miao et al. validation dataset[73] (Fig. 4b); whereas the Clinical Transformer with all features (including clinical and demographics) resulted in a C-index of 0.65 on held-out test splits from Samstein et al. data (Table 2). In contrast to CoxPH and random CoxPH, the Clinical Transformer guided the selection of the functional groups, and the performance could not be recovered using a random selection of functional groups (C-index = 0.55 on test splits from Samstein et al.; HR = 0.76 for Miao et al. dataset[73]) or a random set of 10 hallmark gene sets (C-index = 0.58

on test splits from Samstein et al.; HR = 0.76 for Miao et al[73]. dataset) (Supplementary fig. 6).

Stratifying the population by the median risk score cutoff derived from the discovery dataset (Samstein et al.[33]), the CoxPH model achieved an average HR of 0.45 across testing splits (Fig. 4a). In the pan-cancer IO-treated population from the Miao et al.[73] dataset, the CoxPH model outperformed TMB (median cutoff), achieving an HR of 0.53 (Fig. 4b). However, using non–IO-treated patients in the TCGA dataset, the CoxPH model did not show significant stratification (HR = 0.99). Further evaluation in melanoma datasets showed similar trends. In the Miao et al. dataset[73], the functional-group CoxPH model achieved an HR of 0.52, outperforming TMB (HR = 0.72). The same pattern was observed in the Van Allen et al.[74] dataset, where the CoxPH model achieved an HR of 0.72, outperforming TMB (HR = 0.86; Fig. 4c). Conversely, the opposite trend was observed in the TCGA melanoma dataset (Fig. 4c).

These results demonstrate the potential to translate patterns recognized with complex nonlinear models, such as the Clinical Transformer, into interpretable linear models, such as CoxPH. In addition, our approach enables clearer understanding of feature-outcome relationships and could accelerate the generation of biomarkers insights for clinical development.

### Clinical Transformer embeddings capture biological patterns associated with response

Similar to language models that generate word embeddings encoding both word-level characteristics and contextual semantics[11,75–77], the Clinical Transformer's outcome embeddings encode relationships among the molecular and clinical features in the context of a clinical endpoint (e.g., survival time) that the model was trained to predict. These embeddings may capture higher-order dependencies that unravel biological mechanisms related to response, resistance, or survival. To visualize these relationships, we transformed the 128-dimension outcome embeddings space (Fig. 1b) into a two-dimensional space using the Uniform Manifold Approximation and Projection (UMAP) transformation, a widely used computational biology method[78,79]. By clustering and labeling the outcome embeddings and projecting them onto a UMAP plane, we can explore patterns related to response level, treatment lines, and other clinical outcomes.

Figure 5a presents an example projection for the patients in the Chowell et al.[3] dataset, where patients are organized in a clinically meaningful order on the embeddings plane of the Clinical Transformer. The left panel in Fig. 5a shows a clear trajectory from short-term survivors to long-term survivors for IO-treated patients. The right panel highlights distinct separation of two patient groups, one with prior chemotherapy and the other without.

### Exploring alternatives of clinical response for patients using Clinical Transformer perturbation analysis

To explore alternative responses of short-time survivors, we separately perturbed each of their input features (i.e., artificially changing

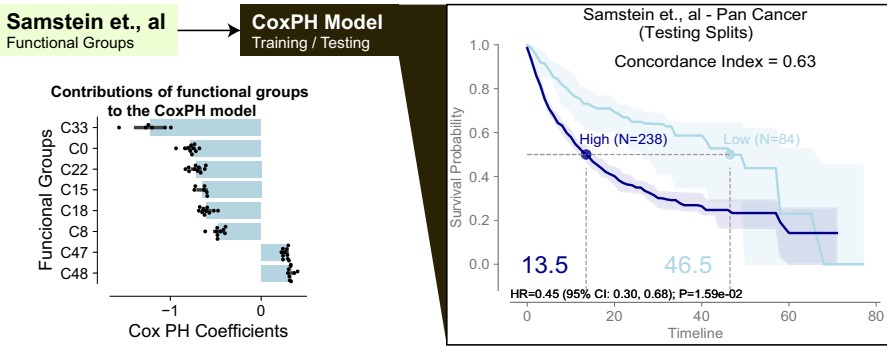

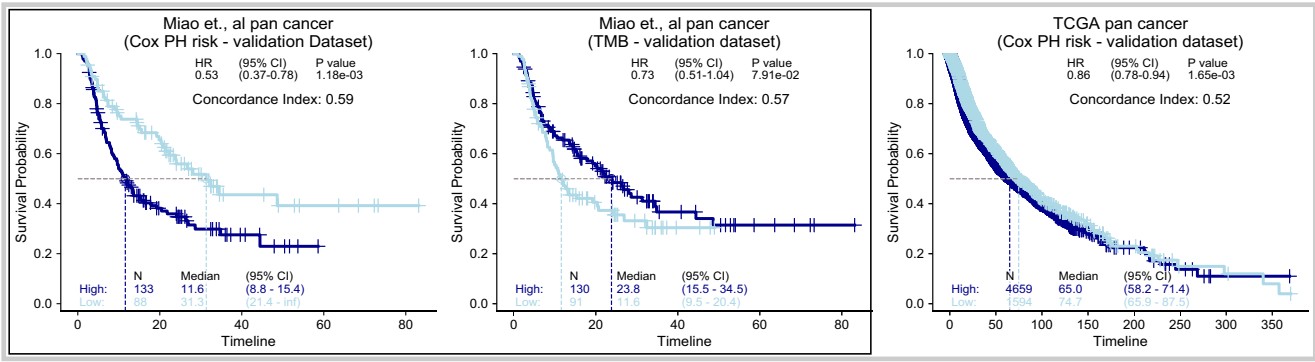

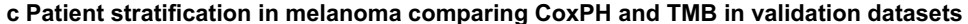

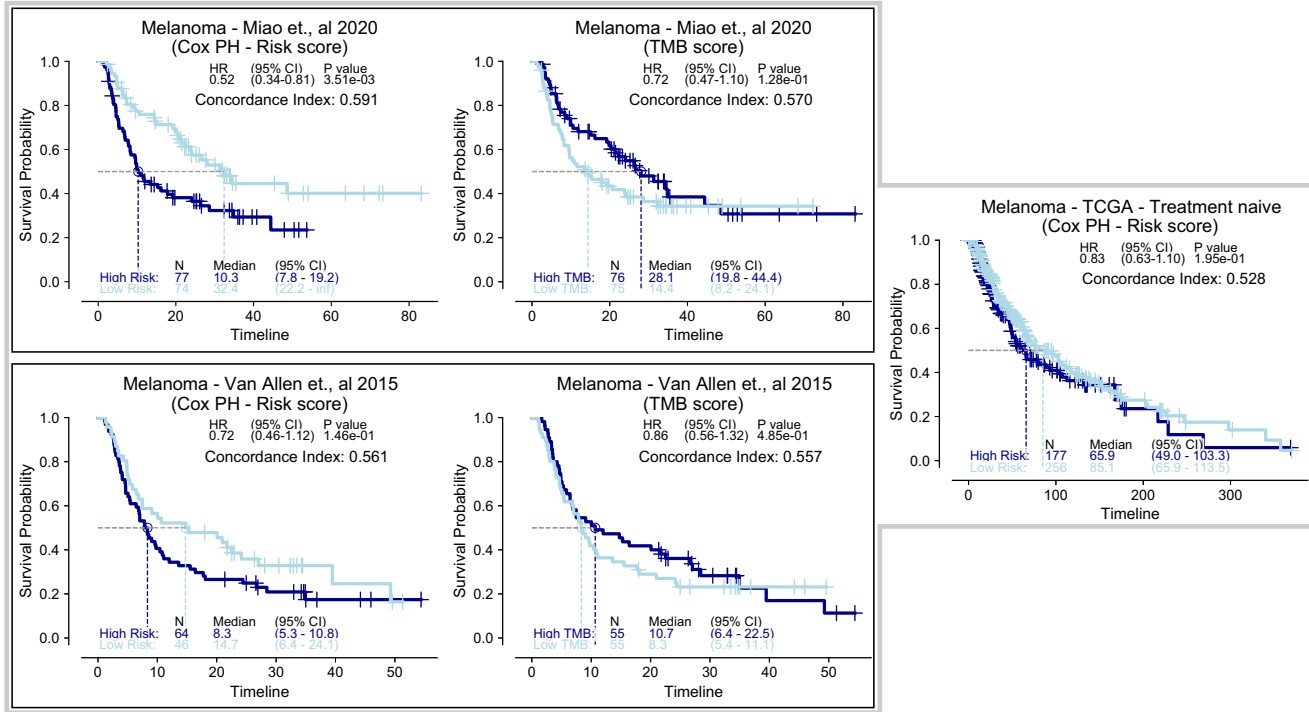

**Fig. 4 | Using top functional groups to yield a simple and interpretable linear model. a** CoxPH model trained on top 10 functional groups derived from Clinical Transformer trained on Samstein et al.[33] discovery dataset. Bar plot (mean) depicts distribution of each CoxPH coefficient across only the train splits where the coefficient *p*-value < 0.05 (Wald test); black dots: individual data points; KM curves of patients from test splits of the 10 models for Samstein et al.[33] discovery dataset, stratified by median training set risk score (partial hazard) from the CoxPH model. **b** KM curves of independent pan-cancer validation datasets, stratified by the CoxPH

(trained on Samstein et al. top 10 functional groups) median training set risk score or by TMB in Miao et al.[73]. IO-treated or TCGA treatment-naïve patients. **c**, KM curves of independent, melanoma-only validation datasets, stratified by CoxPH (trained on Samstein et al. top 10 functional groups) median training set risk score or by TMB in IO-treated (Miao et al., Van Allen et al.), or treatment-naive (TCGA) patients. Shaded region or error bars in (**a**): 95% confidence interval. HR *p*-values in (**b**, **c**) from Wald statistical test. TMB cutoff in (**b**, **c**) = 10 mut/mb. Source data are provided in the SourceData file.

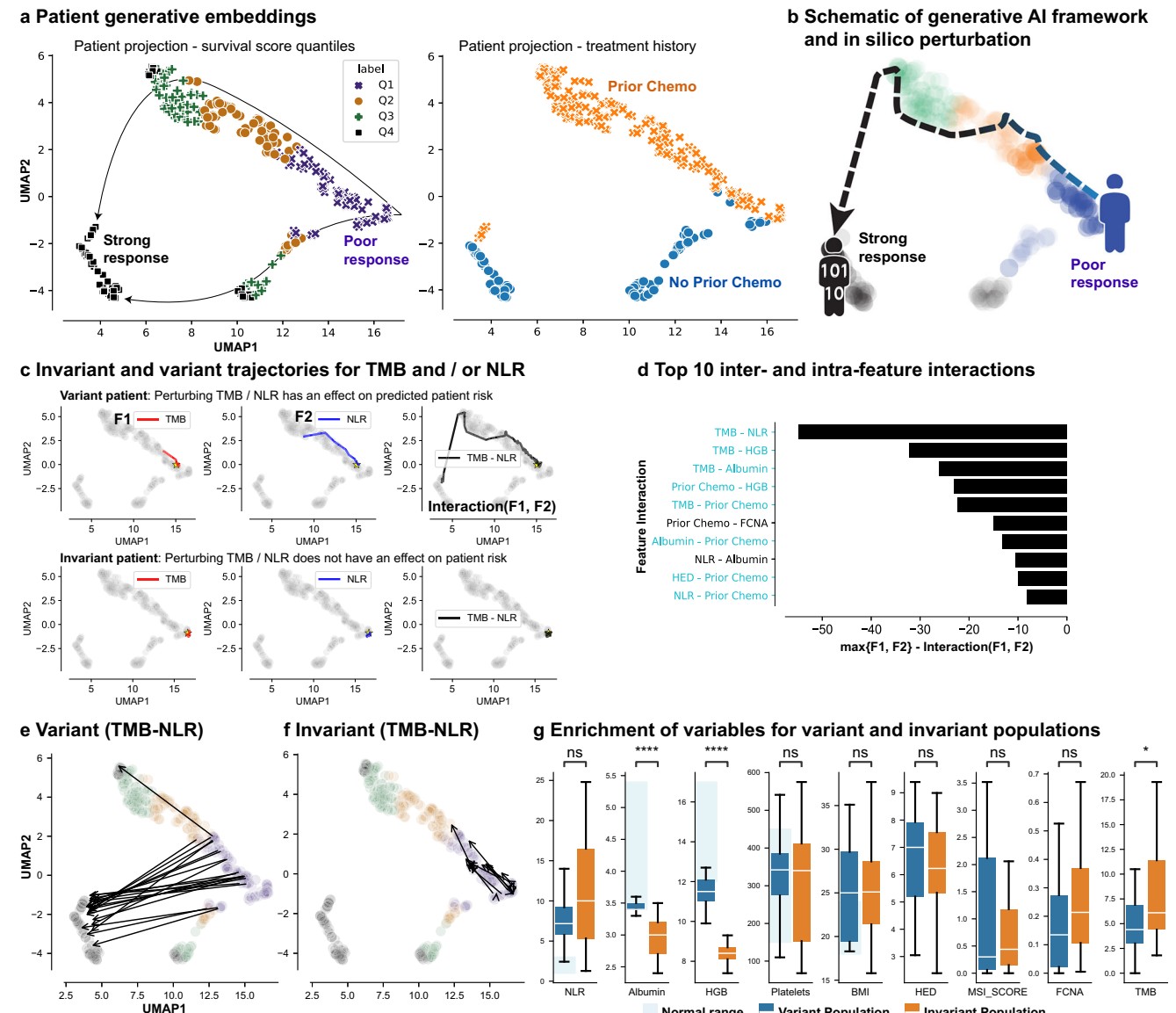

**Fig. 5 | Impact of functional groups on patient response to IO treatment. a** Two-dimensional projection of patient embeddings with UMAP labeled by response of patient populations (left) and by prior chemotherapy ('chemo,' right). Each point in the UMAP plane is a patient's embeddings projection. **b** Schematic of generative AI framework and expected survival trajectory in the latent space of patients under perturbation. **c** (variant patient) Response trajectory for a patient who, under perturbation of the top feature interaction (TMB-NLR), moved from poor-response to super-response. (invariant patient) Response trajectory of patient for whom perturbing TMB-NLR interaction does not affect their risk. Yellow star denotes patient starting location in UMAP plane. **d** Top inter- and intra-feature interactions using the generative module and measuring the impact of perturbing one and two features at a time and calculating the maximum effect of the perturbation for individual and paired features. Interaction values are single point estimates and

shaded bars are used to visually represent their magnitude. Blue interaction pairs: intercluster interactions; black interaction pairs: intracluster interactions. **e** Example of the variant population where patients' survival scores change upon TMB and NLR perturbations. **f** Example of the invariant population where patients' survival scores are mostly unchanged upon TMB and NLR perturbations. **g** Distribution of clinical and molecular features, as compared across the variant ($n = 21$) and invariant populations ($n = 21$). Bonferroni-corrected two-sided $t$ test $P$ value: ns, $5.00e{-}02 < P \le 1.00e + 00$; *, $1.00e{-}02 < P \le 5.00e{-}02$; **, $1.00e{-}03 < P \le 1.00e{-}02$; ***, $1.00e{-}04 < P \le 1.00e{-}03$; ****, $P \le 1.00e{-}04$. Boxplots: center-line, median; box limits, quartile 1 and 3; box whiskers, 1.5x interquartile range; diamonds, outliers; dots, data points. Source data are provided in the SourceData file.

the value of a feature such as NLR, TMB, or gene or pathway expression) while keeping the other features fixed. By comparing the predicted survival score of the patients before and after perturbation (Methods, Supplementary Note 9), we identified patients who may benefit from specific feature modulations related to the therapy's mechanism of action (Fig. 5b). By examining these patients' other features, we may identify new segments for potential treatment. Similarly, understanding patients whose disease failed to improve may inform potential resistance mechanisms and combination opportunities.

Individual feature perturbation (Fig. 5c) showed that patient sensitivity can vary: some were affected by perturbations to one or two features; others showed no sensitivity. Figure 5c (top) shows an example of a patient who was more sensitive to changes when TMB and NLR were perturbed together rather than individually. Figure 5c (bottom) depicts a patient who was not sensitive to NLR or TMB perturbations, individually or together.

Next, we examined all pairwise combinations of features to observe whether any two features influenced survival outcomes. Interestingly, interactions between features from different clusters

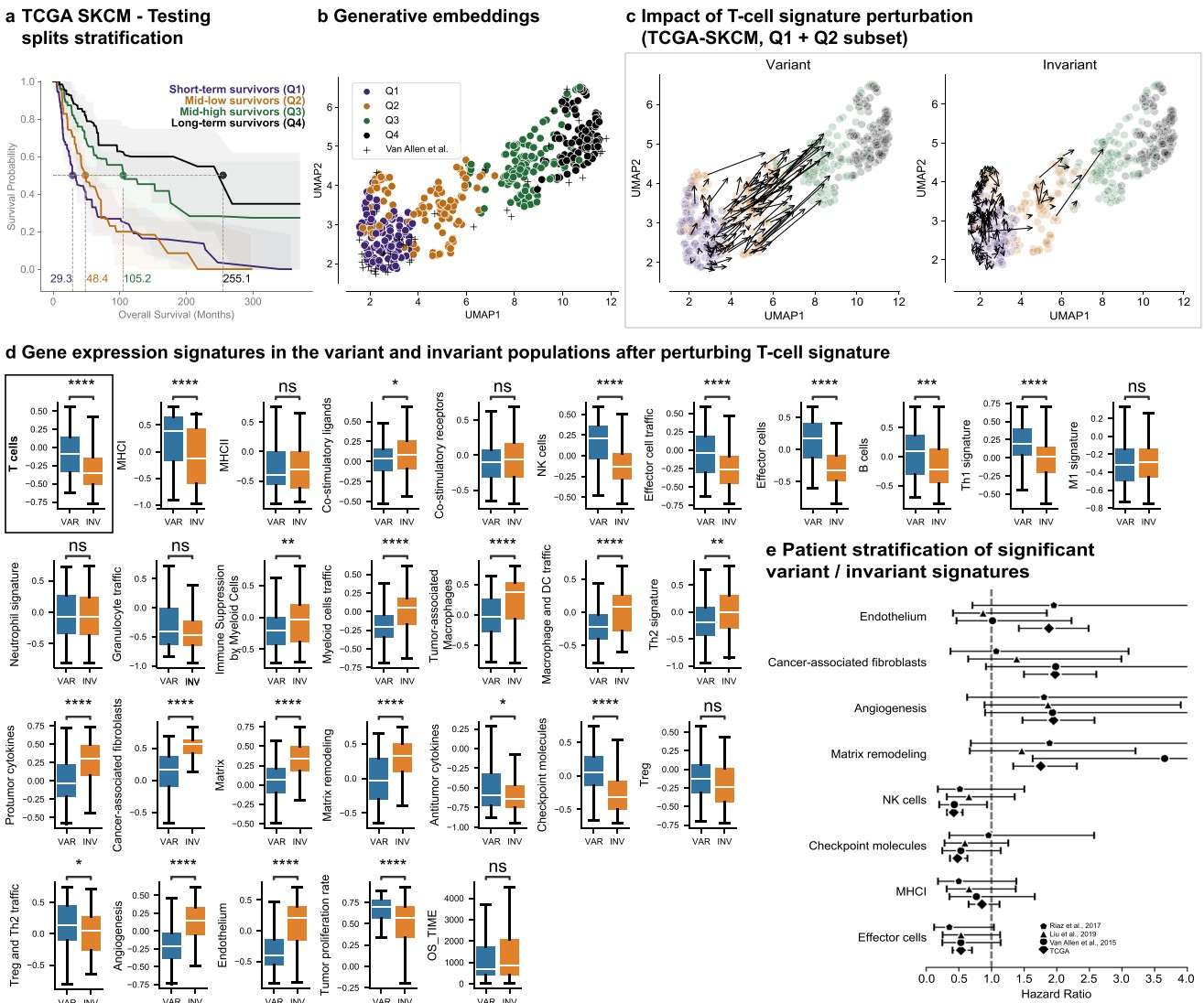

**Fig. 6 | Clinical Transformer and TME on SKCM data. a** Survival stratification of SKCM patients from TCGA. Patients are grouped into quartiles from the Clinical Transformer survival score. The solid line represents the mean survival time across the 10 testing splits. Shaded regions represent the 95% confidence interval. **b** UMAP projection of patient embeddings in the TCGA SKCM dataset. **c** Effect of T-cell perturbations (variant and invariant populations) in TCGA SKCM dataset exclusive to patients in the Q1 and Q2 populations. The size and direction of the arrows reflect the effect and directionality, respectively, of the perturbation. **d** Distribution of TME signatures for the variant (*n* = 115) and invariant (*n* = 115) populations as an effect of T-cell perturbations. Bonferroni-corrected two-sided *t* test *P* value: ns, 5.00e-02 < *P* ≤ 1.00e + 00; *, 1.00e-02 < *P* ≤ 5.00e-02; **, 1.00e-03 < *P* ≤ 1.00e-02; ***, 1.00e-04 < *P* ≤ 1.00e-03; ****, *P* ≤ 1.00e-04. Boxplots: centerline, median; box limits, quartile 1 and 3; box whiskers, 1.5x interquartile range; diamonds, outliers; dots, data points. VAR, variant population; INV, invariant population. **e** CoxPH HR (error bars: 95% CI) of top gene signatures associated with response and resistance in TCGA and IO-treated populations. *N* for each comparison can be found in Supplementary Table 6. Source data are provided in the SourceData file.

(i.e., clusters indicated in Fig. 3d) had a stronger impact on survival than interactions within the same cluster (binomial test between clusters, *P* = 0.004; within clusters, *P* = 0.58; Fig. 5d). TMB-NLR exhibited the most impactful pairwise-feature interaction, followed by TMB-HGB and TMB-albumin (Fig. 5d). This analysis also revealed two population subtypes: a variant population, which exhibited a strong change in survival score when either of those features were perturbed (Fig. 5e), and an invariant population, which was unaffected by perturbations (Fig. 5f). We further investigated patients that exhibited improvement of their predicted survival from short- to long-term (patients labeled Q1 by using the cutoff defined from the training set). As expected, patients from the variant population exhibited values within the normal ranges for albumin, HGB, and platelets (albumin, 3.4–5.4; NLR, 1–3; HGB, 12–17; platelets, 150–450; Fig. 5g, blue boxes). These values may indicate better health compared to patients with values outside of the normal ranges. As expected, high values of TMB

along with low values of NLR tended to produce the greatest changes in survival scores for this population (Fig. 5g). In contrast, the invariant population generally exhibited poorer health, with albumin and HGB outside the normal range (Fig. 5g).

## Identification of potential drivers of response and resistance to immune checkpoint inhibitor treatment via perturbation of a T-cell gene expression signature

We used the Clinical Transformer to train a survival model for melanoma patients (skin cutaneous melanoma [SKCM], IO treatment naive) in TCGA, using tumor microenvironment (TME) gene signatures defined in Bagaev et al.[80]. Quartile survival groups resulting from this training are illustrated in Fig. 6a.

Having established a model of the TME's impact on survival in melanoma patients (Supplementary fig. 7, Supplementary fig. 8; Bagaev et al.[80]), we investigated whether it could help identify drivers

of response and resistance to checkpoint blockade in this setting. Figure 6b presents the UMAP projection of generative embeddings of the patients from this model. To simulate the T-cell infiltration and activation expected during checkpoint blockade, we perturbed the T-cell gene expression signature in Q1 and Q2 patients of the TCGA-SKCM cohort. Similar to the preceding section, this perturbation generated a variant group of patients, where increased T-cell gene expression improved survival, and an invariant group, where it did not. Changes from these perturbations are visualized on the UMAP plane in Fig. 6c, and gene expression signature differences between the variant and invariant groups are presented in Fig. 6d (Supplementary fig. 9). The variant group, which may benefit from checkpoint blockade, demonstrated significantly increased expression of signatures associated with major histocompatibility complex class I (MHC-I) and effector immune cells. This suggests, as previously found[80], that some preexisting immunity to the tumor and functional antigen presentation are key determinants of checkpoint blockade benefit. In contrast, the invariant group, potentially resistant to checkpoint blockade, displayed elevated signatures of angiogenesis, matrix remodeling, and infiltration of cancer-associated macrophages and fibroblasts. These signatures align with known pathways of resistance to checkpoint blockade, such as T-cell suppression by intratumoral macrophages and T-cell exclusion from the microenvironment by cancer-associated fibroblasts and extracellular matrix changes. While the role of angiogenesis in checkpoint blockade response is less well established, its role in blood vessels may impact T-cell access to tumors.

To further validate these findings, we assessed if the identified signatures could stratify melanoma patients treated with checkpoint blockade. Gene signatures were ranked by the magnitude of difference between variant and invariant populations (Supplemental Table 5). The four signatures most associated with the variant population were effector cells, MHC-I, checkpoint molecules, and natural killer cells, while the four signatures least associated were endothelium, cancer-associated fibroblasts, angiogenesis, and matrix remodeling. These eight signatures were examined in three independent studies, selecting patients who received second-line IO treatment and had pre-IO therapy tumor biopsies (Fig. 6e; Supplementary Table 6)[74,81,82]. While survival stratification differences were not statistically significant, several signatures showed consistent trends for differential HRs across studies. For example, high versus low levels of the effector cell signature resulted in HRs of 0.53, 0.53, and 0.35 for survival across the three studies, respectively, whereas matrix remodeling produced HRs of 3.65, 1.46, and 1.89 (Supplementary Table 6).

These results demonstrate the potential for the Clinical Transformer to provide insights into response and resistance, even in early clinical data lacking the treatment under investigation.

## Discussion

Here, we present the Clinical Transformer framework, a deep-learning model with explainability that can predict survival and other clinical outcomes for different lines of treatment with greater accuracy than current state-of-the art prediction models. Unlike other deep neural networks and transformer-based survival models, it explicitly models feature interactions and provides patient-level interpretability. Its transfer learning mechanism allows for self-supervised pretraining on large public datasets like TCGA and GENIE and then fine-tuning on smaller datasets such as those from clinical trials. As a result, the Clinical Transformer was able to make improved survival predictions across several independent datasets, including limited ones for which response and survival end points are available. Across different input modalities, cancer type, and treatments, the Clinical Transformer could capture low- and high-order relationships encoded in clinical, demographic, and molecular data of patients.

We used the cosine-distance between feature embeddings to discover meaningful feature interactions and clusters, such as those separating patient health and immunogenicity. The cosine-distance also helped to map feature dependencies associated with survival and recommend simplified linear models more suitable for clinical practice. For example, cosine-distance clustering identified two functional groups, C47 (HR = 1.32; *APC*, *PIK3CA*, and *TP53*) and C48 (HR = 1.43; *KEAP1* and *STK11*), associated with poor response to immunotherapy. The *KEAP1* and *STK11* genes are widely known for their role in immunotherapy resistance, illustrating the potential clinical utility of feature-based identification of molecular functional groups. These functional groups, when represented as binary variables, enabled the construction of a CoxPH model that outperformed standard methods in predicting patient OS using pan-cancer IO-treated patient datasets. To simplify implementation and interpretation, the integration of functional groups into the CoxPH model was limited to binary variables of a single modality. Future work could explore incorporating groups from multiple modalities and data types by aggregating embeddings for multi-modal functional group constituents into a single score for input into a CoxPH model.

Perturbation analysis demonstrated the Clinical Transformer's utility in early clinical studies lacking treatment data. By simulating molecular changes expected during IO treatment, we identified patients whose survival improved and those whose survival did not. The TME profiles of these patients revealed factors associated with response and resistance to IO treatment, highlighting this approach's potential to identify patients likely to benefit from IO treatment using early data without treatment information.

Despite its strengths, the Clinical Transformer has limitations, as with any transformer model. Computational limitations restrict the model reported in this work to only a few hundred input features, often requiring data aggregation (e.g., RNA gene expression into signatures). Additionally, effective performance on relatively small datasets requires pretraining on thousands of patients with similar sets of features.

We deliberately omitted positional encoding (i.e. the order of the input features) from the Clinical Transformer architecture, despite its importance in enabling transformer models to learn feature context in applications like question-answering. Though existing adaptations of transformers for tabular data include positional encoding[83], biological data from clinical studies is poorly-suited for such approaches. Encoding position in a manner that reflects a biological truth, rather than a (potentially arbitrary) feature order in a data table, is challenging, especially across diverse data types (numerical versus categorical), scales (organism, organ, tissue, cell, or molecular level), and modalities (genomic sequences, demographics, blood tests). Instead, we encoded sets of feature name:value pairs, allowing the model to learn feature context flexibly across different sets. This approach aligns with the objective of positional encoding, while addressing the unique challenges of modeling biological data. Future work should explore novel positional encoding mechanisms tailored to the complexities of biological data.

Unsurprisingly, the performance and interpretability of the Clinical Transformer depends on the quality and perspicuity of the features (e.g., gene expression signatures) it uses. While high prediction performance can be achieved, features lacking direct interpretability (e.g., immune landscape of cancer signatures[34]) generate limited actionable insights. Conversely, well-defined clinical features[80], like those based on melanoma TME, demonstrated better performance. As with other machine learning and AI applications, careful contextual review of the Clinical Transformer's outputs is required to avoid misinterpretation (e.g., prior chemotherapy treatment likely reflects better outcomes in first-line patients rather than a direct association with IO response).

A key assumption of our perturbation analysis is the presence of sufficient natural variation in patient biology and survival outcomes for the model to effectively learn this variation. This depends on the

dimensionality, complexity, and size of the dataset. Additionally, the generative nature of the perturbation analysis may result in "halluci-nations," making validation of findings on independent datasets and careful examination of the results essential.

A key advantage of deep neural networks over conventional machine learning is their flexibility in designing model architecture that can learn from diverse types of datasets. They can use different input features (e.g., molecular signatures, clinical features) and target functions (e.g., prediction of progression-free survival and response). Future work may explore integrating data beyond the non-relational tables used here, enriching learned representations with metadata (e.g. relational tables) or incorporating prior knowledge (e.g. knowl-edge graphs). As preclinical and clinical biological data becomes increasingly available, along with the digitization of clinical samples, and extensive knowledge in the literature, integrating these data to understand diseases is highly valuable. The Clinical Transformer fra-mework presented here is a promising step in this direction.

## Methods

### Clinical datasets and ethics approval
All clinical datasets (except the MYSTIC clinical trial) were from pub-licly available sources, as cited in Table 1. For MYSTIC (NCT02453282), study protocol is available in the original clinical trial publication and was approved by the Institutional Review Boards or Ethics committees of participating cancer treatment centers (203 across 17 countries)[31]. MYSTIC trial data were collected from July 21, 2015 to October 30, 2018; the study was performed in accordance with the Declaration of Helsinki and the International Conference on Harmonization Good Clinical Practice guidelines; and all patients provided written informed consent.

In total, 12 datasets from clinical trials and real-world data (pan-and cancer-specific), totaling 150,070 patients, were used for multiple tasks, including direct, gradual, and transfer learning (Table 1, Sup-plementary fig. 1). Datasets were independently used to predict patient response to treatment in direct and gradual learning modes and combined to assess the impact of transfer learning. All datasets were used to predict treatment response with survival outcomes, except the GENIE project's data, which was used in the self-supervision mode for transfer learning. These data, comprising 134,626 patients[38], informed a gradual learning strategy, using mutations and demographics data as input (Table 1). Instances of this general-purpose model were further fine-tuned on relatively small clinical-trial or real-world datasets to predict patient survival.

### Model architecture
The Clinical Transformer is composed of three main components: (1) an input embedding layer that projects the input features $F \in \mathrm{R}^{N \times 1}$ to an embedding space $P \in \mathrm{R}^{N \times d_k}$ by using the feature name and value pairs; (2) a transformer encoder composed of $l$ number of layers that transforms the input embeddings $E$ into the output embeddings $P \in \mathrm{R}^{N \times d_k}$, which encodes feature interactions via the self-dot product attention; and (3) a prediction layer that uses the output embeddings $P$ from the transformer encoder to perform the training task (survival or self-supervision) via different loss functions.

#### (1) Input embedding layer
The initial transformer architecture proposed by Vaswani et al.[11] uses positional encoding vectors to account for the absolute location of tokens in the sequence. In part because the diverse data types used in the present work lack a universal ordering, we excluded the positional encoding vectors from the transformer architecture. We also validated that this modifica-tion alone led to feature position invariance of the model (Supplementary Note 10; Supplementary fig. 10). The input feature vector $\mathbf{x}$, composed of continuous and categorical fea-tures, is treated as a set of key value pairs $\mathbf{x} \equiv \{\mathbf{x}_k, \mathbf{x}_v\}$, where each element $\{x_i \equiv x_{(i,k)}, x_{(i,v)}\}$ consists of a feature name $k$ and its cor-responding value $v$. To fit into this encoding schema, categorical variables are converted to ordinal arrays (e.g., $n$ categories converted to $n-1$ integers). The input feature vector $\mathbf{x}$ always includes a [CLS] first feature and, if applicable, [PAD] last feature(s).

Feature names $\mathbf{x}_k \in \mathrm{R}^{N \times 1}$ are embedded into a latent repre-sentation $P \in \mathrm{R}^{N \times d_k}$ via a standard text embedding layer, while feature values $\mathbf{x}_v \in \mathrm{R}^{N \times 1}$ are projected into a latent space $P \in \mathrm{R}^{N \times d_k}$ by plugging them into a dense layer. Therefore, each component of the feature value space is decomposed into a linear combination of the learned weights. The aggregated embeddings $E = E_k + E_v$ are the input into the transformer layer (Fig. 1b).

Encoding the input feature names and values enables the model to excel with the type and nature of data commonly found in biological datasets. This embedding schema differs from previous approaches adapting transformer models for tabular data, where inputs are fixed-length vectors, restricted to a fixed (sub)set of features, and may require tabular data-specific modifications of positional encoding to capture context[19–21,83].

Specifically, our input embedding approach enables utili-zation of all the numerous and diverse range of features from different modalities, simultaneous utilization of both dense and sparse features, and utilization of features with any degree of missingness (Supplementary Fig. 11).

Explicitly, we first generate an empirical distribution of number of non-missing features per sample, across all training samples. Using this distribution, we choose a percentile (e.g. 95th percentile) at which we set the hyperparameter corre-sponding to the max length of the input feature vector $\mathbf{x}$. In theory, given infinite compute, this hyperparameter is unne-cessary; in practice, we recommend choosing the smallest value that captures the largest area of this distribution. Note that this max length ≤ max unique features, and that for highly sparse datasets, max length <<max unique features, which allows for reduced model complexity and increased compute efficiency.

Importantly, during each training iteration, for each sample, we randomly sample all non-missing unique features, without replacement, not exceeding the specified max length of $\mathbf{x}$. We then sum across all the token and value embeddings to generate a final set of embeddings for input to the transformer encoder. In all, this allows the Clinical Transformer model to learn across all unique features available in a sample, while disregarding missing values. This also prevents the model from associating the position or ordering of the features with the desired training objective.

#### (2) Transformer encoder
The Clinical Transformer encoder is a multilayer bidirec-tional transformer (similar to BERT) that is based on the original implementation from Vaswani et al.[11] The transformer consists of $l$ blocks containing a multi-head self-attention network, a position-wise feed-forward layer with element-wise addition with a layer normalization. The core of the transformer is the self-attention layer that enables the model to selectively focus on relevant features from the input space by identifying similarities among the input features while associating those similarities with the model outcome. Formally, input embed-dings $E$ are projected into three parametric matrices, the Key ($K$), Query ($Q$), and Value ($V$). Queries represent current information for each input feature, and keys represent the information to which features will be attending. The output of the attention mechanism is defined as the Softmax function of the product

between $Q$ and $K^T$, normalized by a $d_k$ and multiplied by the $V$ matrix as follows:

$$A = \left( \frac{QK^T}{\sqrt{d_k}} \right), \tag{1}$$

$$\text{Attention}(A, V) = \text{Softmax}(A)V \tag{2}$$

where $Q, K, V \in R^{N \times d_k}$. $N$ is the number of input features and $d_k$ is the dimension of key, query, and value vectors. The attention matrix $A \in R^{N \times N}$ captures contextual similarities between queries and keys. The output embeddings of the transformer encoder $P \in R^{N \times d_k}$ are retrieved from the attention matrix after the pointwise and normalization layers.

(3)  Prediction layer

The Clinical Transformer supports two learning modes: (1) time-to-event prediction (survival analysis) to predict patient survival with a given treatment; and (2) unsupervised learning, in which the model identifies feature interactions by looking only at the input features (Fig. 1b), that is, self-supervision. Similar to BERT, we also included the special features [CLS], [MASK], and [PAD] to represent the objective task (survival), masked input features (for unsupervised learning), and unavailable features (for padding absent features), respectively.

## Modeling tasks

**Survival task.** The final hidden state of the special task feature $\mathbf{P}^{[CLS]} \in R^{1 \times d_k}$ is used as the aggregate feature representation of the input data (described as patient embeddings). This vector is passed through a single neuron layer without bias parameter and with weights $W \in R^{1 \times d_k}$. Thus, the survival output score is a scalar defined as $\beta = \mathbf{P}^{[CLS]} * W^T$ with a linear activation function.

To optimize model parameters toward patient survival outcomes, instead of a binary response or text translation, we used the concordance metric in the survival analysis workflow as a measure of model discrimination. Harrell's C-index is defined as the proportion of observations that the model can order correctly in terms of survival times.[84] The C-index can be interpreted as a generalization of the area under the receiver operating characteristic curve that considers censored data. It represents the global encapsulation of the model's discrimination power and its ability to provide a reliable ranking of survival times based on individual risk scores. In our workflow, the concordance-based model discrimination was implemented by using a loss function with a sigmoid approximation of Harrell's C-index. This led to an objective of the form:

$$L_{surv} = \sum_{i,j} w_{i,j} \frac{1}{1 + \exp\left(\frac{\beta_j - \beta_i}{\sigma}\right)} \tag{3}$$

$$w_{ij} = \frac{\Delta_i I(T_i < T_j)}{\sum_{ij} \Delta_i I(T_i < T_j)} \tag{4}$$

where the indices $i$ and $j$ refer to pairs of observations in the data, $\Delta_i = 0$ if censored and 1 if deceased, $T$ is the corresponding survival time, $\beta$ is the predicted survival scores from the Clinical Transformer, and $\sigma$ is a smoothing parameter for the sigmoid approximation.

**Masked pretraining task.** Pretraining a transformer model has proven to be an effective strategy to leverage relevant patterns directly from the raw data in the absence of labels. In the pretraining mode of the Clinical Transformer, 20% of the input feature names are randomly replaced by the special tag [MASK] while its original value is unchanged (similar to language model pretraining). The model is then trained to predict the masked feature names and values by using as input the unmasked features and their respective values (Fig. 1b). The input feature vector is composed of masked and unmasked features and is passed through the Clinical Transformer to obtain the output embeddings $P^{[MASK]} \in R^{F \times d_k}$. These vectors are fed to a dense layer with a Softmax activation function to predict the original masked feature names. The output embeddings $P^{[MASK]}$ are also fed into another dense layer with a linear activation function and outputs the masked feature values. To optimize the model, we used standard categorical cross-entropy for predicting the masked feature names (Loss$_{names}$) and the mean square error loss to predict the masked feature value (Loss$_{values}$). The final loss corresponds to the weighted sum of the independent losses ($\alpha_1$*Loss$_{names}$ + $\alpha_2$*Loss$_{values}$), where $\alpha_1 = 1$ and $\alpha_2 = 0.01$ are set as default parameters and define the contribution of predicting the name and value. In our current settings, we prioritize the prediction of feature names, given that masked values are also included as inputs. The Clinical Transformer framework is also designed to support custom loss functions.

## Model training and evaluation

Motivated by the fact that IO-related clinical data with patient outcomes are difficult to obtain in high numbers, we sought a strategy to enable the use of all available clinical datasets to maximize the value of limited IO clinical data. Our framework can leverage the use of other clinical datasets, even with no outcome labels, to improve predictions in smaller clinical datasets. To this end, we evaluated three training strategies. (1) In direct learning, a machine learning model is trained to predict a target task, given an input dataset. (2) Gradual learning consists of an intermediate step in which a model is trained over unlabeled data taken from the input dataset before training over the target task. In natural-language processing, this is done by predicting a word that is masked in the original text (self-supervision). The obtained model is then transferred to a new model designed to predict the target task of interest. In gradual learning, the same dataset used for pretraining is used for fine-tuning (e.g., pretrain on entire dataset and fine-tune survival on entire dataset). (3) Finally, transfer learning, which is used when a large dataset is available, entails pretraining a large model using all unlabeled data and fine-tuning on a target dataset and task (e.g., pretrain on the GENIE dataset and fine-tune survival prediction over the Samstein et al.[33] dataset). Supplementary Table 7 details parameters and compute utilized for the five pretrained Clinical Transformer models used in the present study.

## Explainability framework

**Permutation feature importance.** To estimate feature importance in the model, we used a feature permutation importance algorithm to perturb variables and compute the difference in the model's C-index output between the unperturbed and perturbed states. Features with stronger effects on outcomes generate larger changes in the model's output. We evaluated feature importance by 10 permutation tests for each trained model over the testing splits.

**Cosine similarity between patient embeddings.** We defined "post-attention" as the pairwise cosine similarity among all features in their latent representations. Formally, for each pair of vectors $\mathbf{P}^{[f_k]}, \mathbf{P}^{[f_l]} \in R^{1 \times d_k}$ from output embeddings, we computed the cosine similarity score as:

$$\cos(f_k, f_l) = \frac{\mathbf{P}^{[f_k]} \cdot \mathbf{P}^{[f_l]}}{\left\| \mathbf{P}^{[f_k]} \right\| \left\| \mathbf{P}^{[f_l]} \right\|} \tag{5}$$

where $f_k$ and $f_l$ are any two features present in the input data point $x_i$. The complete pairwise feature similarities can be depicted as the

square matrix $S(x_i) \in R^{F \times F}$. The rationale for using the output embeddings from the last encoder layer is that those embeddings preceded the outcome classifier networks (survival, masked prediction). These embeddings represent the aggregated information of the input feature interactions in the sample $x_i$.

Given that the outcome embedding vector $\mathbf{P}^{[CLS]} \in R^{1 \times d_k}$ is directly associated with the outcome variable (because this vector is used as input to the prediction layer), the cosine similarity between this vector and any feature vector $\mathbf{P}^{[fk]}$ reflects the contribution of the feature $fk$ to the outcome. Thus, for a given input data point $x_i$, we can identify and rank the features, encoding the similar information to the predicted outcome by computing all their cosine similarity scores to the outcome embedding.

Similarly, the cosine similarity among feature embeddings $\mathbf{P}^{[fk]}$, $\mathbf{P}^{[fl]} \in R^{1 \times d_k}$, where $f_l, f_k \neq$ [CLS], depicts local relationships and feature interactions. Similar to language models, in which two words can describe semantic similarities within a certain context, the cosine similarity between two features describes their information content and can be seen as a local feature interaction.

**Functional groups.** A functional group $\zeta$ is defined as the collection of interacting features encoding related information on a global scale. To obtain functional groups, we averaged the pairwise similarities $S(x_i) \in R^{F \times F}$ across a given population (e.g., an entire population or specific populations) and clustered them, using agglomerative clustering (for Samstein et al. data), k-means (for Chowell et al. data), or any clustering algorithm that can better represent the data, with a predefined number of $\zeta_n$ clusters. For selecting the best number of clusters, the elbow or silhouette score was applied. To perform clustering analysis, missing pairs are encoded with 1 value. See Supplementary Note 11 in SI for more information.

**Functional group ranking.** In a specific patient subpopulation $c \in C$, where $C$ represents all patient groups (e.g., labels in a classification problem or response population in survival analysis), the aggregated mean cosine similarity of a functional group $\zeta_c$ over the subpopulation $c$ represents the effect of the functional group in the subpopulation. Therefore, mean cosine similarities close to 1 indicate that the functional group is characteristic of the subpopulation, whereas a mean cosine score close to 0 reflects high orthogonality, indicating that the functional group is not associated with the given subpopulation. Core functional groups are those that show a high cosine similarity across all subpopulations, whereas target functional groups are enriched on specific subpopulations. Therefore, we ranked the functional groups by their mean cosine similarity on each subpopulation to describe their most informative functional groups.

**Validation of mutational functional groups.** The mutational data offer a direct alternative to validate the effect of functional groups from the input data. In particular, we can measure the impact of mutations in patient survival outcome by binarizing each functional group to either mutation or wild-type status. Formally, a functional group is defined as a collection of mutated genes, $\zeta_k = \{g_1, \ldots, g_i, \ldots, g_{Gk}\}$, where $G_k$ represents the genes in the group $k$. Therefore, for a given patient $I$, we can measure whether the functional group $\zeta$ is mutated if at least one gene in $G_k$ is mutated as follows:

$$\bar{\zeta}_{i,k} = \begin{cases} 1 & if \ \sum_{g}^{G} x_{i,g} > 0 \\ 0 & other \end{cases} \quad (6)$$

where $k$ represents the $k$th functional group and $x_{i,g}$ represents the gene value $g$ for patient $i$. We then fit a univariate CoxPH model for each binary functional group, selected only those groups that are statistically significant ($P < 0.05$), and ranked them based on their

HR. Statistically significant functional groups were tested on the complete training set and on a fully independent validation dataset.

**Functional group simplification of survival modeling.** A subset of functional groups are strongly associated with patient survival. To maximize the signal from the functional groups, we trained a multivariate CoxPH regression model using the top 10 binarized functional groups $\zeta$ as input. This model uses a subset of input features (aggregated genes as functional groups), reducing the model's complexity. To evaluate the performance of the model, we used 10 training and testing splits of 80% and 20%, respectively, over the Samstein et al.[33] pan-cancer dataset. To evaluate the model on independent datasets, we trained a CoxPH model using the entire Samstein et al.[33] dataset. The model outperformed TMB and achieved comparable performance to the Clinical Transformer model trained with all input features (genes). The multi-CoxPH model was also compared with a control model in which random functional groups $\zeta^{rand}$ were built by randomly selecting the same number of genes as the real functional groups $\zeta$ to train a multivariate CoxPH model. The multivariate CoxPH model was evaluated in the pan-cancer and cancer-specific settings across multiple studies and trials.

### Clinical Transformer as generative model

To extract potential trajectories in the single-variable perturbation setting, we perturbed the input feature $f_i$ by sampling the feature according to its distribution in the training population (i.e., divided the distribution of feature $f_i$ across all patients into 10 percentiles and used its corresponding value as perturbation). This generated a new set of output embeddings along with their corresponding survival scores. On the other hand, to measure the impact of two feature interactions (pairwise interactions), we randomly sampled each pair of feature values from the training data while keeping all other features constant. We repeated this process 50 times for each feature pair and each patient. We were then able to identify the features and interactions with the strongest impact on the model's output (survival score).

### Feature-feature interactions from the generative model for patient survival scores

Let $P$ be the population of patients and let $F$ be the set of features. For each patient $p$ in $P$, we performed $F$ single simulations by perturbing one feature at a time, and $F \times F$ paired simulations by perturbing two features at a time.

For each simulation, we recorded the patient's maximum survival score. Let $M_p(f)$ denote the maximum survival score for patient $p$ in the simulation where feature $f$ is perturbed, and let $M_p(f, g)$ denote the maximum survival score for patient $p$ in the simulation where features $f$ and $g$ are perturbed. For each feature $f$, we have a distribution of maximum survival scores, consisting of $M_p(f)$ for all patients in $P$. Similarly, for each feature pair $(f, g)$, we have a distribution of maximum survival scores, consisting of $M_p(f, g)$ for all patients in $P$.

We computed the Mann-Whitney nonparametric statistical test between the distribution of maximum survival scores for each feature and feature pair and the distribution of the survival scores from the original data (without perturbation). Let $P_{val}(f)$ and $P_{val}(f, g)$ be the $P$ values resulting from the statistical test for feature $f$ and feature pair $(f, g)$, respectively. We transformed $P_{val}(f)$ and $P_{val}(f, g)$ to the $-\log$ scale, denoted as $-\log(P_{val}(f))$ and $-\log(P_{val}(f, g))$, respectively. To evaluate the effect of an interaction being stronger than the individual counterparts, we used the following equation:

$$\text{Diff} = \max\{-\log(P_{val}(f)), -\log(P_{val}(g))\} - (-\log(P_{val}(f, g))) \quad (7)$$

where $P_{val}(f)$ and $P_{val}(g)$ are the $-\log$ of the $P$ values for the features $f$ and $g$, respectively, and $P_{val}(f, g)$ is the $P$ value of the interaction between features $f$ and $g$. If Diff is negative, it represents an

aggregated value for the interaction between features $f$ and $g$, indicating that the combined perturbation of both features is more significant than the individual perturbations. If Diff is positive, it indicates that the individual perturbations are enough to improve patient survival score.

## Analysis of variant and invariant populations

A feature or set of features $F$ is perturbed $n$ times, generating a conditional output patient embedding $P^{[CLS]}(M|F = f_i)$, as well as a conditional survival score $\beta(M|F = f_i)$, where $M$ represents the trained Clinical Transformer model and $f_i$ a given perturbation $i<n$ in the feature $f$. We can define the change in survival scores for a given patient $j$ by taking the difference between the maximum and minimum of the perturbed survival scores $\beta_j^f = \{\beta_{j1}^f, \ldots, \beta_{jn}^f\}$ over the feature (or set of features). The $\Delta\beta_j^f = \max\{\beta_j^f\} - \min\{\beta_j^f\}$ describes the impact of the perturbation of the feature $f$ in the patient survival score, indicating the sensitivity of the patient $j$ to a given perturbation $f$. We can then define two populations based on the median of the distribution of all patients $\Delta\beta_j^f$: a variant population defining patients with a $\Delta\beta_j^f$ score higher than the median of the $\Delta\beta_j^f$ distribution, and an invariant population defining patients with a $\Delta\beta_j^f$ below the median of the $\Delta\beta_j^f$ distribution. In the perturbation analysis for the Chowell et al.[3] and Bagaev et al.[80] datasets, we defined the variant/invariant populations exclusively on the poor and low-mid survivors in order to identify patients that under a perturbation show a transition that reflects improved survival.

## Statistical analysis and modeling

For time-to-event survival data, hazard ratios and their 95% confidence intervals were computed by fitting a univariate Cox proportional hazards model (lifelines Python package). *P*-values for hazard ratios were computed with a Wald test. For all other univariate comparisons of continuous distributions, a two-sided statistical test was used (two-sample independent Student's *t*-test or Mann-Whitney-Wilcoxon test). All statistical tests were corrected for multiple comparisons, where appropriate. All survival times are reported in months.

Analyses were performed and models built in Python (3.8.10) using the following packages: flaml==1.0.11, lifelines==0.27.1, numpy==1.22.4, pandas==1.4.3, scikit-learn==0.24.1, scipy==1.9.0, tensorflow==2.10.0rc1, UMAP-learn==0.5.3.

## Reporting summary

Further information on research design is available in the Nature Portfolio Reporting Summary linked to this article.

# Data availability

All datasets (except for the MYSTIC clinical trial) are publicly available at the sources listed in Table 1. For MYSTIC, the data are available under restricted access to fulfill legal and ethical obligations regarding patient data. AstraZeneca group of companies allows researchers to submit a request to access anonymized patient-level clinical data, aggregated clinical or genomics data (when available), and/or anonymized clinical study documents through the Vivli (https://vivli.org) web-based data request platform, in accordance with AstraZeneca's clinical data access policy: https://www.astrazenecaclinicaltrials.com/our-transparency-commitments. An independent Scientific Review Board will review requests and time to request fulfillment may take up to a year. Source data are provided with this paper.

# Code availability

The code used in this study is available at: https://doi.org/10.24433/CO.6684503.v1

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

## Acknowledgements

We thank Diego Chowell for fruitful discussion about the manuscript, Gary Doherty for comments and feedback on the draft, Elizabeth Choe for rephrasing and edits, and Deborah Shuman and Joey Stewart for help with the edits

## Author contributions

G.A.-A., conceptualization, methodology, software, validation, data curation, visualization, formal analysis, investigation, writing - original draft, reviewing, and editing E.K., methodology, software R.S., resources, validation G.J.S., writing – reviewing and editing A.P., writing – original draft I.K., investigation E.J., conceptualization, methodology, supervision, investigation, validation, writing - original draft, reviewing, and editing

## Competing interests

All authors are current or former employees of AstraZeneca with stock ownership, interests, and/or options in the company.
