## [Transparent Peer Review file · Nature Communications]

Pretrained transformers applied to clinical studies improve predictions of treatment efficacy and associated biomarkers

Corresponding Author: Dr Etai Jacob

Version 0:

Reviewer comments:

Reviewer #1

(Remarks to the Author)

The author successfully developed and validated a pretrained transformer model Clinical Transformer. The model can be applied to clinical or molecular data to predict treatment efficacy via patient survival. The model also aims to improve interpretation and identify important biomarkers via functional groups obtained by grouping similar features.

The authors compared the proposed clinical transformation model with Cox-PH, random survival forest and biomarkers(BMT) but not other deep learning based algorithms. Many previous studies have proposed deep learning models to predict patient survival using similar data, from basic DL models such as DeepSurv and Cox-nnet to other transformer-based survival prediction models. A comparison with deep learning based models is necessary. The authors clustered features from the clinical transformer into functional groups using cosine similarity. What clustering algorithms were used on each dataset in the study? How does the author evaluate the clustering result of functional groups? What metrics were used? It's recommended that the authors elaborate on the detailed clustering process and clustering performance.

It seems a simplified Cox-PH model based on functional groups is only done on the IO dataset, possibly because it's easier to aggregate features if they are binary. Is it possible to build functional group-based models on clinical data or mixed data types? How do functional groups benefit the clinical translation of clinical data or mixed data?

The author applied the functional group based cox-PH model on multiple validation datasets and reported their performance(fig 3h, 3i). In addition to HR and log-rank p-values of the dichotomized risk group, the authors should also report the c-index of Cox-PH risk score and patient survival on each validation dataset. C-index can better measure the prediction power of a survival model and is most used in survival modeling.

How should one interpret the biological meaning of functional clusters? Have the authors tried to link derived functional clusters to existing literature? Do they confirm or conflict with previous findings on cancer survival?

Please elaborate on the data preprocessing steps before fitting the model. Did the authors remove any samples or modify the data in any way? How does the clinical transformer model handle missing values or sparse features?

Minor points

The author applied different learning tasks on multiple datasets. It will help the readers if the author can demonstrate exactly what task is applied on which dataset with a workflow illustration.

In figure3E, the author is simultaneously testing multiple functional groups, adjusted p-value instead of p-value should be reported

Line 179 c-index should be 0.73,

Reviewer #2

(Remarks to the Author)

The manuscript introduces a transformer-based deep learning framework for predicting survival outcomes in cancer treatment. Due to my background and expertise, I am neither able to evaluate the significance of this work in the area of clinical medicine, nor assess the quality of its results regarding clinical data. On the other hand, I have some concerns from the perspective of transformer-based models and would like to propose three suggestions.

1.
The manuscript presents an evaluation of the model against state-of-the-art methods in survival prediction. However, it lacks the comparison with:

a. More traditional deep learning (DL) approaches commonly used for tabular data, such as Multi-Layer Perceptron (MLP). This would provide a more comprehensive understanding of the advantages and necessity of employing transformers in clinical prediction. If there are existing contributions that have applied these approaches in clinical prediction, this literature should be introduced, compared, and cited in the manuscript.

b. Other transformer models specifically developed for tabular data, such as TableFormer (see: 10.18653/v1/2022.acl-long.40). It would be beneficial to include a comparison, or at least a discussion, on how the clinical transformer compares or contrasts with these approaches, particularly regarding positional encoding, as other contributions employed modified positional encoding approaches for tabular data instead of simply removing it (refer to Badaro et al. for a review: 10.1162/tacl_a_00544).

I understand the authors' approach offers significant flexibility, notably in training with small datasets and handling missing data. While these are innovative and valuable aspects of this research, it would be advantageous to assess whether this flexibility results in any compromise in performance compared to other DL- and transformer-based models.

2.
The manuscript mentions randomizing the order of input features to enforce the model to ignore positions during training (line 714 to 720). However, as positional encoding is removed, the positions of features shouldn't affect the model's performance and this enforcement is unnecessary. To validate that the positions do not affect the model, an investigation into whether altering the order affects the result can be conducted, i.e., if changing the order of the input features used for prediction, will the model output exactly the same result?

The reliability and consistency of prediction models, I believe, are paramount in clinical applications, and thus I suggest the authors add this validation, especially since the authors seem unconfident with the sufficiency of removing positional encoding.

3.
The authors claim that the clinical transformer is a foundation model for clinical applications (line 105). Foundation models refer to generalizable models that can be adapted to a wide range of downstream tasks. However, the manuscript only introduced and validated the clinical transformer as a survival prediction approach. I wonder what other clinical tasks it can be adapted to. Additionally, the potential inclusion and treatment of metadata, such as relational tables, in the model can further extend its generalizability. Clarifying how such data is accommodated within the proposed approach will strengthen the understanding of its applicability in diverse clinical scenarios.

In conclusion, from the perspective of transformer-based model, the study presents a promising framework with the potential to significantly improve the performance, generalizability, and flexibility in clinical prediction based on tabular data. Addressing these points will not only reinforce the credibility of your findings but also broaden the scope of its application in real-world clinical settings. I look forward to seeing the revised manuscript with these considerations.

Version 1:

Reviewer comments:

Reviewer #2

(Remarks to the Author)

The authors' responses and added validations are satisfactory. One additional point regarding the positional encoder: the authors mention that "it is unclear how to encode position in a manner that reflects a biological truth...". This could potentially be investigated by comparing different positional encoder mechanisms proposed by AI researchers, not just the existing implementations in the clinical field. Exploring which existing positional encoders represent the current state-of-the-art for the authors' proposed clinical transformer could provide valuable insights. The authors may consider adding this discussion as future work.

(Remarks on code availability)

Upon reviewing the code, the authors provided a toy example of the implementation of the clinical transformer instead of the original model studied in the manuscript. This is understandable, as the original model may be very large and computationally expensive to run. If this is the case, the authors are suggested to elaborate on some training details such as the model size, GPU configurations, training time, etc.

The toy example presented in the code is based on a synthetic dataset of 10 features, labeled from f_0 to f_9. What do these features represent? Additionally, the example uses the same 10 features throughout, without demonstrating the flexibility of the clinical transformer in handling missing data, which is a significant advantage of the proposed approach. A demonstration of how the model manages missing data would be beneficial.

Reviewer #4

(Remarks to the Author)

The author designs a new transformer-based method to predict survival outcomes in cancer treatment. However, the author's responses are not sufficient solve the reviewer 1's concerns, and the innovation remains unclear. The benchmark section does not convincingly demonstrate the superior performance of the proposed tool. There are many mismatched and inconsistent results between the figures and their descriptions. Besides Reviewer 1's comments, there are also new concerns regarding the model design.

Major:

- The benchmarking results didn't show significant improvement compared to existing deep learning-based models. Please justify the rationale for the new model design for treatment efficacy. Additionally, how were the hyperparameters selected in the tool's comparison?
- The functional groups selected and evaluated by the same method need to be justified. Silhouette score is a clustering method evaluation method. How do you elucidate the groups related to functions?
- Linking the groups and functions should be validated by pathway enrichment.
- Please justify that the predicted missing values are not false positives.
- This framework effectively handles relatively small datasets by incorporating a transfer learning mechanism. However, the results on small datasets rely on the model parameters learned from large available datasets. In this paper, TCGA and GENIE are selected as the pre-training datasets. The authors should describe the criteria for selecting these two datasets.
- The authors use a feature permutation importance algorithm to estimate feature importance and identify features associated with survival outcomes. However, in a transformer-based framework, the attention matrix is typically used to describe the importance of each feature pair. The authors should explain why they did not consider using the attention matrix.
- Based on suggestions 2 and 3, the necessity of the transformer needs to be justified. If the goal is merely to capture complex, nonlinear relationships among the features, other methods like convolutional neural networks (CNNs) can accomplish this task as well.
- The authors used the top 10 binarized functional groups to train a multivariate Cox proportional hazards regression model. They should explain the criteria for selecting the cutoff of the top 10. Additionally, during the analysis of translating the clinical transformer to simple and interpretable linear models, some functional groups in the top 10 have adjusted P values greater than 0.1. It is not reasonable to select functional groups that are not statistically significant.

Minor

- Lines 442 and 443: The hazard ratio (HR) should be 0.45, and (Fig. 3h) should be 0.53 (Fig. 3e).
- The distribution of figures in Figure 3 is confusing, especially Figure 3h and Figure 3i. Additionally, the caption of Figure 3 does not correspond to the content of Figure 3.

(Remarks on code availability)

Version 2:

Reviewer comments:

Reviewer #4

(Remarks to the Author)

The author has addressed all my comments.

(Remarks on code availability)

REVIEWER COMMENTS AND AUTHORS' RESPONSES

Reviewer #1 (Remarks to the Author):

The author successfully developed and validated a pretrained transformer model Clinical Transformer. The model can be applied to clinical or molecular data to predict treatment efficacy via patient survival. The model also aims to improve interpretation and identify important biomarkers via functional groups obtained by grouping similar features.

Referee's point:

The authors compared the proposed clinical transformation model with Cox-PH, random survival forest and biomarkers (BMT) but not other deep learning based algorithms. Many previous studies have proposed deep learning models to predict patient survival using similar data, from basic DL models such as DeepSurv and Cox-nnet to other transformer-based survival prediction models. A comparison with deep learning based models is necessary.

Answer:

We have evaluated five additional neural network models (Transformer, Cox-nnet, DeepSurv, Neural MTLR, and NNET-Survival) for predicting survival time to event outcomes as follows. Detailed description of methods in Supplementary Material under A8 Benchmarking against other neural network architectures section. Overall, the clinical transformer outperformed all the other neural network architectures – Table 2 in main text is updated with all the additional comparisons.

Referee's point:

The authors clustered features from the clinical transformer into functional groups using cosine similarity. What clustering algorithms were used on each dataset in the study? How does the author evaluate the clustering result of functional groups? What metrics were used? It's recommended that the authors elaborate on the detailed clustering process and clustering performance.

Answer:

We have added the following text to the SI with additional notes in the main text directing to this section:

"A12 Choosing the number of clusters (functional groups)

For the identification of the number of functional groups from the cosine similarity scores, we used a conventional silhouette score-based methodology (arbitrary, other method could have been chosen). For instance, in the Chowell et al., dataset, we used the k-means clustering algorithm across a predefined range of potential cluster numbers (2 to 10). The silhouette score for each predefined number of clusters was computed to quantitatively assess the clustering quality. This metric evaluates the degree of similarity of an instance to its assigned cluster in relation to other clusters. The selection of the optimal number of clusters was defined by the highest silhouette score (in this case k=4). This approach ensures

that the chosen cluster solution maximizes intra-cluster similarity while maintaining clear delineation between different clusters.”

Referee’s point:

It seems a simplified Cox-PH model based on functional groups is only done on the IO dataset, possibly because it’s easier to aggregate features if they are binary. Is it possible to build functional group-based models on clinical data or mixed data types? How do functional groups benefit the clinical translation of clinical data or mixed data?

Answer:

We have indeed created a simplified CoxPH model based on functional groups within a single modality and on binary data due to the simplicity in implementation and clarity of the interpretation (applied to Samstein et., al dataset). We agree with the reviewer that utilizing functional groups from different data types in a simplified model is desirable, although can be challenging and currently beyond the scope of this study. We have nonetheless added a passage to the discussion detailing ideas and considerations as follows:

“Although for simplicity of implementation and interpretation, the integration of functional groups into a Cox PH model was limited to binary variables of a single modality, future work could explore how to utilize groups from multiple modalities and data types. For instance, one could aggregate embeddings for multi-modal functional group constituents into a single score (e.g. by computing the magnitude of their vector sum) to be used as an input for a Cox PH model. “

Referee’s point:

The author applied the functional group based cox-PH model on multiple validation datasets and reported their performance (fig 3h, 3i). In addition to HR and log-rank p-values of the dichotomized risk group, the authors should also report the c-index of Cox-PH risk score and patient survival on each validation dataset. C-index can better measure the prediction power of a survival model and is most used in survival modeling.

Answer:

We agree. Concordance index added to the figures.

Referee's point:

How should one interpret the biological meaning of functional clusters? Have the authors tried to link derived functional clusters to existing literature? Do they confirm or conflict with previous findings on cancer survival?

Answer:

We agree that these are important point to emphasize in the manuscript. Therefore, we have added to the main text several sentences further explaining the functional clusters and linking them to existing literature with references as follows:

- Regarding TMB and MSI: "Both MSI and TMB have been associated with improved outcome following treatment with immunotherapy, forming the basis for the tumor agnostic approval of anti-PD-1 in patients with either of these markers^{9, 50}, while HLA-LOH has been previously associated with resistance to immunotherapy⁵¹."
- Regarding other features: "Cluster 2 consisted of patient-related variables, including albumin, body mass index, HED, HGB, NLR, platelets, and age. Many of these markers are reflective of overall patient health and inflammatory status, and have previously been associated with differential benefit from immunotherapy, though the extent to which they are predictive vs prognostic is not clear⁵⁶⁻⁵⁹."

We have also added to the main text a paragraph indicating that a resulted cluster is less consistent in the literature:

- "Two functional groups, denoted C47 (HR = 1.32; *APC*, *PIK3CA*, and *TP53*) and C48 (HR = 1.43; *KEAP1* and *STK11*), were associated with short-term survival with immunotherapy. In line with these findings, multiple studies have shown the negative prognostic impact of *KEAP1* and *STK11* for both immunotherapy and chemotherapy⁶²⁻⁶⁵, and *while the impact of mutations in APC, PIK3CA and P53 is less consistent⁶⁶⁻⁷¹, these are all critical driver genes in several tumors, whose impact might be expected to vary based on the specific setting and therapeutic regimen.*"

Referee's point:

Please elaborate on the data preprocessing steps before fitting the model. Did the authors remove any samples or modify the data in any way?

Answer:

Some data types such as molecular (RNA, Mutations) needed a pre-processing before usage. We included the descriptions of such transformations in the supplementary file section A9 - Data Preprocessing. Most of the data used in this study was already structured for machine learning usage from their respective publication.

Referee's point:

How does the clinical transformer model handle missing values or sparse features?

Answer:

We have added the following paragraphs to the main text to explain this important attribute of the clinical transformer:

“Specifically, our input embedding approach enables utilization of all the numerous and diverse range of features from different modalities, simultaneous utilization of both dense and sparse features, and utilization of features with any degree of missingness (Supplementary Fig. SF.12).

Explicitly, we first generate an empirical distribution of number of non-missing features per sample, across all training samples. Using this distribution, we choose a percentile (e.g. 95th percentile) at which we set the hyperparameter corresponding to the max length of the input feature vector x . In theory, given infinite compute, this hyperparameter is unnecessary; in practice, we recommend choosing the smallest value that captures the largest area of this distribution. Note that this max length \leq max unique features, and that for highly sparse datasets, max length \ll max unique features, which allows for reduced model complexity and increased compute efficiency.

Importantly, during each training iteration, for each sample, we randomly sample all non-missing unique features, without replacement, not exceeding the specified max length of x . We then sum across all the token and value embeddings to generate a final set of embeddings for input to the transformer encoder. In all, this allows the clinical transformer model to learn across all unique features available in a sample, while disregarding missing values. This also prevents the model from associating the position or ordering of the features with the desired training objective. “

Minor points

Referee’s point:

The author applied different learning tasks on multiple datasets. It will help the readers if the author can demonstrate exactly what task is applied on which dataset with a workflow illustration.

Answer:

We agree, therefore added to Supplementary Information **Supplementary Figure SF.10.**

Referee's point:

In figure 3E, the author is simultaneously testing multiple functional groups, adjusted p-value instead of p-value should be reported

Answer:

P-values were adjusted using Sidak method and updated to the figure.

Referee's point:

Line 179 c-index should be 0.73,

Answer:

The error is fixed in the main text.

Reviewer #2 (Remarks to the Author):

The manuscript introduces a transformer-based deep learning framework for predicting survival outcomes in cancer treatment. Due to my background and expertise, I am neither able to evaluate the significance of this work in the area of clinical medicine, nor assess the quality of its results regarding clinical data. On the other hand, I have some concerns from the perspective of transformer-based models and would like to propose three suggestions.

1.

The manuscript presents an evaluation of the model against state-of-the-art methods in survival prediction. However, it lacks the comparison with:

Referee's point:

a. More traditional deep learning (DL) approaches commonly used for tabular data, such as Multi-Layer Perceptron (MLP). This would provide a more comprehensive understanding of the advantages and necessity of employing transformers in clinical prediction. If there are existing contributions that have applied these approaches in clinical prediction, this literature should be introduced, compared, and cited in the manuscript.

Answer:

We have evaluated five additional neural network models (Transformer, Cox-nnet, DeepSurv, Neural MTLR, and NNET-Survival) for predicting survival time to event outcomes as follows. Detailed description of methods in Supplementary Material under A8 Benchmarking against other neural network architectures section. In overall, the clinical transformer outperformed all the other neural network architectures – these additional analysis is updated in Table 2 in the main text.

Referee's point:

b. Other transformer models specifically developed for tabular data, such as TableFormer (see:

10.18653/v1/2022.acl-long.40). It would be beneficial to include a comparison, or at least a discussion, on how the clinical transformer compares or contrasts with these approaches, particularly regarding positional encoding, as other contributions employed modified positional encoding approaches for tabular data instead of simply removing it (refer to Badaro et al. for a review: 10.1162/tacl_a_00544).

Answer:

We appreciate this point and added the following section to the main text to address it: “We have deliberately omitted positional encoding (i.e. the order of the input features) from the clinical transformer architecture, despite its critical role in enabling any transformer-based model to learn feature context necessary for, e.g. question-answer applications. Indeed, existing adaptations of transformers for tabular data employ some derivative of positional encoding⁸³. However, biological data from clinical studies is not well-suited to simple positional encoding. It is unclear how to encode position in a manner that reflects a biological truth, rather than a (potentially arbitrary) choice of feature ordering in a data table – and in a way that is valid across diverse data types (numerical versus categorical), scales (organism, organ, tissue, cell, or molecular level), and modalities (genomic sequences, demographics, blood tests). We therefore adopted an approach that instead encodes sets of feature name:value pairs, such that the model can still learn the feature context (and across different sets) in a more flexible manner. Our approach is thus consistent with the objective of positional encoding, while also being well-adapted to the challenges inherent in modeling biological data.”

Referee’s point:

I understand the authors’ approach offers significant flexibility, notably in training with small datasets and handling missing data. While these are innovative and valuable aspects of this research, it would be advantageous to assess whether this flexibility results in any compromise in performance compared to other DL- and transformer-based models.

Answer:

We appreciate the reviewer’s recognition of the significant flexibility offered by our approach, particularly in training with small datasets and handling missing data. While these aspects are indeed innovative and valuable, we understand the importance of assessing whether this flexibility compromises performance compared to other DL- and transformer-based models.

In our revised manuscript, we present comprehensive benchmarking results demonstrating that the clinical transformer outperformed all other five neural network architectures. This thorough evaluation showcases that our framework’s flexibility does not come at the expense of performance. For detailed insights into the performance comparison between our method and other DL- and transformer-based models, we invite the reviewer to refer to our response to reviewer point 1 above and the revised table 2 in the main text).

Referee’s point:

2.

The manuscript mentions randomizing the order of input features to enforce the model to ignore

positions during training (line 714 to 720). However, as positional encoding is removed, the positions of features shouldn't affect the model's performance and this enforcement is unnecessary. To validate that the positions do not affect the model, an investigation into whether altering the order affects the result can be conducted, i.e., if changing the order of the input features used for prediction, will the model output exactly the same result?

The reliability and consistency of prediction models, I believe, are paramount in clinical applications, and thus I suggest the authors add this validation, especially since the authors seem unconfident with the sufficiency of removing positional encoding.

Answer:

We acknowledge the reviewer's suggestion regarding the removal of positional encoding and its expected impact on the transformer's sensitivity to feature order. To address this concern, we conducted a shuffling experiment, the details of which are provided in the Supplementary Information (section A11 and figure SF.11) as follows:

“To confirm that the clinical transformer, which lacks position encoding, is not sensitive to feature order, we performed the following experiment. We trained a modified version of a clinical transformer model on data from Chowell et al. that lacks feature order shuffling (i.e. samples features in a constant order). During model inference, we shuffled the order of features and generated 10 shuffled replicates for each sample. We then computed the standard deviation of model predictions for each sample against their 10 replicates. The standard deviations showed an exceedingly low variance (below $1e-6$, see Figure SF.11 below). This result provides clear empirical evidence that feature order does not affect the predictive consistency of the clinical transformer.”

Additionally, we would like to highlight in the context of the feature order the issue of handling sparse, high-dimensional input sets, exemplified by the GENIE dataset discussed in the manuscript. Despite comprising 2290 features, the dataset exhibits an average of only 6 non-null features per sample. In such cases, utilizing the entire feature space for model input becomes impractical due to the excessive padding required. Therefore, we adopted a strategy of utilizing only the available features for a given sample, analogous to the approach used in natural language processing (NLP), where the feature set represents the vocabulary, and each sample corresponds to a short sentence.

Referee's point:

3.

The authors claim that the clinical transformer is a foundation model for clinical applications (line 105). Foundation models refer to generalizable models that can be adapted to a wide range of downstream tasks. However, the manuscript only introduced and validated the clinical transformer as a survival prediction approach. I wonder what other clinical tasks it can be adapted to.

Answer:

We appreciate the suggestion of the reviewer as we believe that this has strengthened our work and demonstrates the general applicability of the clinical transformer framework. We have refined the phrasing of line 105 and added an additional section to the main text and SI (indicated below), that supports the potential of the clinical transformer to serve as a framework to build foundation models that can be adapted for a wide range of applications.

“Utility of the clinical transformer beyond survival prediction

We assessed the adaptability of the clinical transformer beyond its initial application in survival prediction, postulating its potential as a foundational model for various clinical tasks. Specifically, our objective was to discern whether the model could accurately classify the origin of cancer samples being metastatic or primary, exclusively in patients with established metastatic disease.

Therefore, we downloaded prostate cancer data from the MSK-MeTropism dataset, comprising 2,172 samples, with 1,762 from metastatic disease (939 primary and 823 metastatic samples). To build our foundation model, we used the GENIE v15 dataset, excluding samples associated with the MSK-MeTropism project. Thus, we trained our foundation model on 167,421 of the 198,041 total samples in GENIE for 10,000 iterations. We did not use our pre-trained GENIE v11 foundation model, as it already included the sample type as an input feature along with MSK-MeTropism samples, precluding its use for this task. We selected all available molecular features aggregated at the gene level, counting only missense and frameshift variants (copy number variants were already extracted at the gene level). Additionally, we included patient age, sex, race, cancer type, sample type, cancer type detailed, sample type detailed, and sequencing center for training the GENIE v15 foundation model. We evaluated the clinical transformer against random forest model using different proportions of data for training. We split the data into train/test splits of 80% and 20%, respectively and repeated the process 10 times and reported the AUC-ROC on the test set. The clinical transformer with fine-tuning outperformed the random forest model across all training data fractions (Supplementary table ST.13). The advantage was most pronounced with small training sizes; using only 5% of data for training, the clinical transformer achieved an AUC of 0.761 (± 0.011) compared to 0.722 (± 0.017) for random forest.

This analysis underscores the versatility of the clinical transformer beyond survival prediction, highlighting its potential to serve as a foundational model adaptable to various downstream clinical tasks.”

Referee’s point:

Additionally, the potential inclusion and treatment of metadata, such as relational tables, in the model can further extend its generalizability. Clarifying how such data is accommodated within the proposed approach will strengthen the understanding of its applicability in diverse clinical scenarios.

Answer:

This work is currently focused on handling input feature space organized in a single tabular format. However, we are exploring ways to incorporate prior knowledge into the model, including relationships between different features within tables (e.g., knowledge graphs) or scales of data (including both single cell and bulk data in one model). For example, we are considering the use of knowledge graphs to integrate interactions between input features (e.g., genes, pathways) via input embeddings or as

conditional attentions such as in TableFormer. Such implementations can enhance the ability of the clinical transformer to leverage large and diverse datasets (e.g., pre-clinical data) and incorporate prior knowledge (e.g., gene-gene interactions or literature), thereby improving further performance for different clinical applications. We have added the following note to the discussion accordingly:

“Future work may explore how to integrate data beyond the non-relational tables used in the present study, to enrich learned representations with metadata (e.g. as inherent to relational tables) or to include prior knowledge (e.g. with knowledge graphs).”

Referee:

In conclusion, from the perspective of transformer-based model, the study presents a promising framework with the potential to significantly improve the performance, generalizability, and flexibility in clinical prediction based on tabular data. Addressing these points will not only reinforce the credibility of your findings but also broaden the scope of its application in real-world clinical settings. I look forward to seeing the revised manuscript with these considerations.

REVIEWER COMMENTS

Reviewer #2 (Remarks to the Author):

The authors' responses and added validations are satisfactory. One additional point regarding the positional encoder: the authors mention that "it is unclear how to encode position in a manner that reflects a biological truth...". This could potentially be investigated by comparing different positional encoder mechanisms proposed by AI researchers, not just the existing implementations in the clinical field. Exploring which existing positional encoders represent the current state-of-the-art for the authors' proposed clinical transformer could provide valuable insights. The authors may consider adding this discussion as future work.

We have added this to our discussion as follows

Future work should explore and investigate how novel positional encoding mechanisms might be developed and used in a manner best suited for such biological data modeling challenges.

Reviewer #2 (Remarks on code availability):

Upon reviewing the code, the authors provided a toy example of the implementation of the clinical transformer instead of the original model studied in the manuscript. This is understandable, as the original model may be very large and computationally expensive to run. If this is the case, the authors are suggested to elaborate on some training details such as the model size, GPU configurations, training time, etc.

Indeed, the provided code with a toy dataset was released to demonstrate, in a tractable manner, the capabilities of the clinical transformer including pre-training, fine-tuning, feature importance, interpretability. We have added a new supplementary table (ST.14) that details the model architecture, number of parameters, GPU used for training, and the approximate training time for the 5 pretrained clinical transformer models used in our study (SF.10). We have now also provided the source code used for the experiments performed in the present manuscript. The capsule is updated and approved by codeocean for peer review. Upon publication, we will release the clinical transformer as an API that can be easily used, integrated, and extended to other projects or applications to ensure its adaptability.

The toy example presented in the code is based on a synthetic dataset of 10 features, labeled from f_0 to f_9 . What do these features represent? Additionally, the example uses the same 10 features throughout, without demonstrating the flexibility of the clinical transformer in handling missing data, which is a significant advantage of the proposed approach. A demonstration of how the model manages missing data would be beneficial.

The synthetic dataset contains 2 features with signal that relates to the time-to-event outcome (f_0, f_1) and 8 features with random noise (f_2, f_3, \dots, f_9). We have now

added a vignette to the example that demonstrates how the model can manage missing data.

Reviewer #4 (Remarks to the Author):

The author designs a new transformer-based method to predict survival outcomes in cancer treatment. However, the author's responses are not sufficient solve the reviewer 1's concerns, and the innovation remains unclear. The benchmark section does not convincingly demonstrate the superior performance of the proposed tool. There are many mismatched and inconsistent results between the figures and their descriptions. Besides Reviewer 1's comments, there are also new concerns regarding the model design.

In our responses below, we have clarified the innovation of our work and how our benchmarks convincingly demonstrate superior performance of our proposed tool. We have also fixed any mismatched or inconsistent results between the figures and their descriptions.

Major:

1. The benchmarking results didn't show significant improvement compared to existing deep learning-based models. Please justify the rationale for the new model design for treatment efficacy. Additionally, how were the hyperparameters selected in the tool's comparison?

The clinical transformer clearly outperformed all other deep learning-based models (5 in total) across all different benchmark datasets (5 in total; Table 2), as assessed by the concordance index (c-index) metric, and with a low standard deviation, indicating that the c-indices were meaningfully separated. These other deep learning-based models included those leveraging multi-layer perceptrons (Cox-nnet, DeepSurv, Neural MTLR), those utilizing convolutional neural networks (Nnet-survival), and transformers (Transformer Survival). The clinical transformer outperformed each other method by an overall average of nearly 10%. This supports our notion that use of our clinical transformer is justified. In addition, the explainability module and perturbation analysis enables better understanding of the efficacy signal, increased confidence in the model and translational capability to clinical settings.

We did not perform hyperparameter tuning for the clinical transformer. We had previously indicated in Supplementary Information: **A8 Benchmarking against other neural network architectures**, that for comparator methods,

We used default parameters for most of the architectures, except for Neural MTLR where the learning rate was set to 0.0001 because of exploding gradients, and for Transformer survival model where we selected the best model based on the plateau of the test loss

2. The functional groups selected and evaluated by the same method need to be

justified. Silhouette score is a clustering method evaluation method. How do you elucidate the groups related to functions?

Please see our combined response to concerns 2 & 3 in our response to concern 3.

3. Linking the groups and functions should be validated by pathway enrichment.

We had previously defined functional groups in our Methods section (Explainability framework – Functional groups). We have now also added a more explicit description of a functional group in our main text as follows (underlined)

Input features from Chowell et al. data were then grouped into four “functional groups” based on their cosine similarity (Methods). We define a functional group as a collection of interacting features that encode related information on a global scale – in this example, the functional groups shared similar information with respect to the survival endpoint (Fig. 3d)

In other words, a “functional group” derived from a clinical transformer model is a potentially *de novo* collection of features – including those that span modalities, like those in the Chowell et al. dataset analysis (e.g. molecular like TMB, blood labs like albumin, and demographic, like age). Some functional groups, like those that are derived exclusively from genomic data in the Samstein et al. dataset analysis, may map to known biological ontologies, such as gene sets representing biological pathways. Indeed, for the Samstein et al. dataset analysis, we performed a GO enrichment analysis, which yielded only one significant pathway association (GO:0002682; see section: **Identification of key functional groups associated with survival outcomes**). That other pathways were insignificant, and that the clinical transformer can create functional groups comprising of multiple modalities, highlights the fact that the clinical transformer may be identifying new facets of biology. These may require future experimental elucidation.

The process of deriving a functional group follows a standard procedure for clustering in the machine learning community: build a model to perform clustering, and in the absence of ground truth labels, use the model itself to assess the quality of clustering. The model with the highest quality (in this case, the highest Silhouette score) is then used to output the final clustering – i.e. the functional groups. As mentioned above, the biological interpretation (‘function’) of these groups may or may not be known. The resulting functional groups one derives is therefore also dependent on how the clustering was performed.

4. Please justify that the predicted missing values are not false positives.

The clinical transformer does not predict missing values. Per our Methods **(1) Input embedding layer**, the clinical transformer is designed in a way that “allows the clinical transformer model to learn across all unique features available in a sample, while disregarding missing values”. In other words, nowhere does the model predict missing values; rather, the model can flexibly handle input data with missing values.

5. This framework effectively handles relatively small datasets by incorporating a transfer learning mechanism. However, the results on small datasets rely on the model parameters learned from large available datasets. In this paper, TCGA and GENIE are selected as the pre-training datasets. The authors should describe the criteria for selecting these two datasets.

Indeed, by definition, transfer learning requires pretraining on large datasets. We selected two of the most well-established, large-scale, highly cited datasets in the field of oncology. GENIE “is a publicly accessible cancer registry of real-world clinic-genomic data assembled through data sharing between 19 leading international cancer centers” (<https://www.aacr.org/professionals/research/aacr-project-genie/>) and developed under the aegis of the premiere professional association for cancer research, the American Association for Cancer Research (AACR). It has been cited over 1300 times. TCGA “molecularly characterized over 20,000 primary cancer and matched normal samples spanning 33 cancer types”, has “generated over 2.5 petabytes of genomic, epigenomic, transcriptomic, and proteomic data”, and is a “joint effort between NCI and the National Human Genome Research Institute” (<https://www.cancer.gov/tcga>). The original TCGA papers were cited nearly 6000 times.

These two data resources are therefore very well-established and contain the thousands of patients’ molecular and clinical features required to train a model that can successfully be used for transfer learning.

6. The authors use a feature permutation importance algorithm to estimate feature importance and identify features associated with survival outcomes. However, in a transformer-based framework, the attention matrix is typically used to describe the importance of each feature pair. The authors should explain why they did not consider using the attention matrix.

We had previously indicated the limitations of using the attention matrix, in the section entitled “Identification of key functional groups associated with survival outcomes”,

The attention mechanism has been previously employed to interpret model predictions. For instance, in Xie et al., Vashishth et al., and Vig and Belinkov, attention weights were used as maps to investigate salient interactions among features. A limitation of these studies was that they considered only the attention network’s weights, which may not provide a direct interpretation and can be influenced by nonlinear relationships within other transformer components (e.g., feed-forward network following multi-head attention, layer normalization). Because these weights may not necessarily correlate with importance values derived from gradient-based methods, they do not fully capture the underlying significance of the features (Jain et al., Rigotti et al., and Chefer et al.).

It has been suggested that attention heads within a transformer model specialize in learning distinct aspects of the input data. However, we lack a universal method for summarizing information from multiple attention heads. Simply averaging or aggregating the attention weights from these diverse heads would lead to loss of information.

This is why, as a first pass, we employed the well-established method of feature permutation importance, to assess global feature importance. And then, as a key novel contribution of our work, we detailed our “post-attention” explainability module, which is based upon cosine similarity between model embeddings (which are contextualized representations of input that are downstream of the transformer block components), and which avoids the aforementioned limitations of utilizing the attention matrix.

7. Based on suggestions 2 and 3, the necessity of the transformer needs to be justified. If the goal is merely to capture complex, nonlinear relationships among the features, other methods like convolutional neural networks (CNNs) can accomplish this task as well.

The clinical transformer clearly outperformed all other deep learning-based models (5 in total) across all different benchmark datasets (5 in total; Table 2), as assessed by the concordance index (c-index) metric, and with a low standard deviation, indicating that the c-indices were meaningfully separated. These other deep learning-based models included those utilizing convolutional neural networks (Nnet-survival). The clinical transformer exceeded the CNN performance across all benchmarks by an average of ~10%. Other deep learning methods tested included multi-layer perceptrons (Cox-nnet, DeepSurv, Neural MTLR) and transformers (Transformer Survival).

Overall, the clinical transformer outperformed all 5 deep learning methods by an overall average of nearly 10%. This supports our notion that use of our clinical transformer is justified.

8. The authors used the top 10 binarized functional groups to train a multivariate Cox proportional hazards regression model. They should explain the criteria for selecting the cutoff of the top 10. Additionally, during the analysis of translating the clinical transformer to simple and interpretable linear models, some functional groups in the top 10 have adjusted P values greater than 0.1. It is not reasonable to select functional groups that are not statistically significant.

By design of our framework, all functional groups associate with survival outcomes (see section: **Identification of key functional groups associated with survival outcomes**). Some may associate to a stronger or weaker degree. The example in Fig. 3e-f illustrates how one could isolate which functional groups most strongly relate to survival outcomes, so as to utilize only a subset for a simple and interpretable linear model. Across the univariate Cox regressions, we most care about the rank ordering of the effect size (i.e. the hazard ratios) and the rank ordering of the statistical significance (Fig. 3e). In other words, this can be seen as a feature selection step (if we treat

functional groups as input features to some downstream linear model). We have removed p-values from Fig. 3e to avoid confusion.

Our choice of top 10 was the most parsimonious, logical choice. In the figures for reviewer-only below, we show how a Cox PH model trained on any number from 1 through 29 functional groups relates to survival (via concordance index; trained on Samstein et al. with 80/20 train/test split for 10 replicates as in Fig. 3). Across the cross-validated training data (Samstein et al.) and an independent validation set (Miao et al.), it is clear that performance of a model that utilizes a subset of top-ranked functional groups clearly outperforms models using a collection of hallmark gene sets (see section: **Translating the clinical transformer to simple and interpretable linear models**) or models using a random selection of functional groups. Future work can examine more sophisticated functional group ‘feature selection’ methods; it is up to future users of our framework to decide what subset of functional groups is most informative for their downstream applications.

A cox PH model trained using the top n functional groups (Samstein et al. dataset), and evaluated on test splits from Samstein et al., shows improved concordance index performance until a plateau of >10 functional groups, consistent with what one might expect if there was a truly generalizable signal:

A cox PH model trained using n random functional groups (Samstein et al. dataset), and evaluated on test splits from Samstein et al., shows a gradual increase in correlation with concordance index, consistent with what one would expect – increase performance as the model is complexity increases using features selected randomly:

A cox PH model trained using the top n functional groups (Samstein et al. dataset), and evaluated on the independent Miao et al. dataset, shows improved concordance index performance similar to that seen on the training data, consistent with the notion that choosing top 10 functional groups generalizes well.

In contrast, a cox PH model trained using n random functional groups (Samstein et al. dataset), and evaluated on the independent Miao et al. dataset, shows no correlation with concordance index, supporting the notion that taking random functional groups will fail to generalize.

Minor

9. Lines 442 and 443: The hazard ratio (HR) should be 0.45, and (Fig. 3h) should be 0.53 (Fig. 3e).

We agree that the original description in lines 442-443 may have been unclear. We have revised it as follows

The Cox PH model resulted in an average C-index of 0.63 on held-out test splits from Samstein et al. data (Fig. 3g), with an HR of 0.53 on the Miao et al. validation data set⁷⁵; whereas the clinical transformer with the whole set of features (including clinical and demographics) resulted in a C-index of 0.65 on held-out test splits from Samstein et al. data (Table 2).

10. The distribution of figures in Figure 3 is confusing, especially Figure 3h and Figure 3i. Additionally, the caption of Figure 3 does not correspond to the content of Figure 3.

We have modified the layout of Figure 3 to better highlight the grouping of plots for each panel. To avoid confusion, we moved the description of boxplots to the description of panel **c**, rather than after the description of panel **i**.